EMBO
Molecular Medicine

# CARD14 signaling in intestinal epithelial cells induces intestinal inflammation and intestinal transit delay

Aigerim Aidarova[1,2], Marieke Carels [1,2], Mira Haegman [1,2], Yasmine Driege[1,2], Steven Timmermans[2,3], Eline Van Damme [1,2,4], Javier Aguilera-Lizarraga[5,12], Maria Francesca Viola [5], Rita de Cássia Collaço [6], Joan Manils[7,8], Steven C Ley [9], Frank Bosmans [6,10], Tom Van de Wiele[4,11], Guy Boeckxstaens[5], Claude Libert [2,3], Rudi Beyaert [1,2,13]✉ & Inna S Afonina [1,2,13]✉

## Abstract

CARD14 is an intracellular NF-κB signaling mediator in the skin, and rare CARD14 variants have been associated with psoriasis and atopic dermatitis. CARD14 is also expressed in intestinal epithelial cells (IEC). However, its function in the intestine remains unknown. We demonstrate here that transgenic mice expressing the psoriasis-associated gain-of-function human *CARD14*(E138A) mutant specifically in IEC show mild intestinal inflammation, without epithelial damage. Moreover, *CARD14*(E138A)[IEC] mice show a drastic reduction in intestinal motility, often associated with rectal prolapse. Enteric neuronal survival and functionality are unaffected in *CARD14*(E138A)[IEC] mice. Transcriptome analysis of IEC from *CARD14*(E138A)[IEC] mice reveals decreased expression of antimicrobial peptides by Paneth cells, accompanied by microbial dysbiosis and increased susceptibility to enteric bacterial infection. Our findings suggest that gain-of-function CARD14 mutations may not only predispose patients to psoriasis but also mild intestinal inflammation, reduced intestinal motility, and increased sensitivity to intestinal infection. *CARD14*(E138A)[IEC] mice are also a valuable tool for further investigation of IEC-intrinsic molecular processes involved in intestinal inflammation and motility disorders.

**Keywords** CARD14 Signaling; Inflammation; Intestinal Motility; Paneth Cells; MALT1
**Subject Categories** Digestive System; Genetics, Gene Therapy & Genetic Disease

## Introduction

Caspase recruitment domain family member 14 (CARD14) is an intracellular signaling protein that is expressed in skin and mucosal tissues (Jordan et al, 2012a; Fuchs-Telem et al, 2012; Manils et al, 2020). Several gain-of-function *CARD14* mutations have been found to be associated with psoriasis in humans and murine models (Jordan et al, 2012b; Van Nuffel et al, 2020), while loss-of-function mutation has been associated with atopic dermatitis (Peled et al, 2019). Mice expressing hyperactive *CARD14* mutants exhibit psoriasis-like dermatitis characterized by keratinocyte proliferation, epidermal acanthosis, hyperkeratosis, immune cell infiltration and increased expression of inflammatory cytokine/chemokines (Mellett et al, 2018; Van Nuffel et al, 2020; Manils et al, 2020) which is partially dependent on IL-17A and IL-23 (Wang et al, 2018; Mellett et al, 2018).

CARD14, together with CARD9, CARD10, and CARD11, belongs to the CARD-CC protein family, which is defined by an evolutionarily conserved caspase activation and recruitment domain (CARD) and a coiled-coil (CC) domain (Staal et al, 2018). Initially identified in placenta (Bertin et al, 2001), *CARD14* expression was subsequently reported in various cell types, including epithelial cells of skin and esophagus, Langerhans cells, γδ T cells (Fuchs-Telem et al, 2012), dermal endothelial cells (Harden et al, 2014), and epithelial cells of the intestine (Manils et al, 2020; Wittner et al, 2023). Together with BCL10 (B-cell lymphoma/leukemia 10) and the paracaspase MALT1 (mucosa-associated lymphoid tissue lymphoma translocation protein 1), CARD14 forms a so-called CBM signaling complex that activates NF-κB (nuclear factor-kappa B) and MAPK (mitogen-activated protein kinase) signaling, leading to the expression of several genes involved in cell proliferation, differentiation, and inflammation (Afonina et al, 2016; Howes et al, 2016). The identity of upstream

[1]Center for Inflammation Research, Unit of Molecular Signal Transduction in Inflammation, VIB, Ghent, Belgium. [2]Department of Biomedical Molecular Biology, Ghent University, Ghent, Belgium. [3]Center for Inflammation Research, Unit of Mouse Genetics and Inflammation, VIB, Ghent, Belgium. [4]Center for Microbial Ecology and Technology (CMET), Faculty of Bioscience Engineering, Ghent University, Ghent, Belgium. [5]Center of Intestinal Neuroimmune Interaction, Translational Research Center for Gastrointestinal Disorders (TARGID), KU Leuven, Leuven, Belgium. [6]Department of Basic and Applied Medical Sciences, Ghent University, Ghent, Belgium. [7]Immunity, Inflammation and Cancer Group, Oncobell Program, Bellvitge Biomedical Research Institute, Barcelona, Spain. [8]Serra Húnter Programme, Immunology Unit, Department of Pathology and Experimental Therapy, School of Medicine, University of Barcelona, Barcelona, Spain. [9]Institute of Immunity & Transplantation, University College London, London NW3 2PP, UK. [10]Department of Pharmaceutical Chemistry, Drug Analysis, and Drug Information, Research Group Experimental Pharmacology, Center for Neurosciences, Vrije Universiteit Brussel, Brussels, Belgium. [11]International Associated Labs Homigut University of Clermont-Auvergne/Ghent University, Ghent, Belgium. [12]Present address: Institute for Neurosciences, CSIC–University Miguel Hernández (UMH), Sant Joan d'Alacant, Spain. [13]These authors contributed equally to this work as senior authors: Rudi Beyaert, Inna S Afonina. ✉E-mail: rudi.beyaert@irc.vib-ugent.be; inna.afonina@irc.vib-ugent.be

activators of CARD14 is still largely unclear. Although Toll-like receptor 3, Dectin 1, and IL-17 have been proposed to activate CBM signaling in keratinocytes, these observations still await independent verifications (Schmitt et al, 2016; Wang et al, 2018; Mazzone et al, 2020). Therefore, functional studies of CARD14 signaling predominantly rely on the ectopic expression of *CARD14* psoriasis-associated mutants in cellular and animal models. Hyperactivation of *CARD14* specifically in keratinocytes is sufficient for the induction of the rapid development of psoriasis-like skin inflammation in mice (Van Nuffel et al, 2020; Manils et al, 2020; Zhang et al, 2021b).

Apart from the observation that *CARD14* expression is increased in ulcerative colitis patients and positively correlates with disease severity (Yamamoto-Furusho et al, 2018; Wittner et al, 2023), its functional role in the gut is largely uncharacterized. Here, we describe the generation and characterization of mice expressing a human gain-of-function *CARD14*(E138A) transgene specifically in intestinal epithelial cells (IEC). *CARD14*(E138A) mice show mild intestinal inflammation in the intestinal tract. Unexpectedly, *CARD14*(E138A) mice also suffer from a drastic decrease in intestinal motility. Moreover, we show that *CARD14*(E138A) signaling in IEC not only induces inflammatory gene expression, but also decreases expression of antimicrobial peptides (AMP) by Paneth cells, which is accompanied by reduced microbial diversity and increased susceptibility to enteric bacterial infection.

# Results

## *CARD14*(E138A) expression in IEC induces mild intestinal inflammation

We first studied *Card14* expression in the intestine of wild-type mice by RNAscope. In agreement with previous reports (Manils et al, 2020; Wittner et al, 2023), *Card14* was expressed in IEC of both small intestine and colon (Fig. 1A). To study the effect of CARD14 signaling in IEC, we generated mice expressing *CARD14*(E138A), a rare gain-of-function variant that was originally described in psoriasis patients, specifically in IEC (referred as *CARD14*(E138A)$^{IEC}$ mice). Expression of *CARD14*(E138A) in IEC was confirmed in the small intestine and colon via qPCR and western blotting (Fig. EV1A,B). *CARD14*(E138A)$^{IEC}$ mice were born at normal Mendelian ratios and did not show any macroscopically visible signs of disease, such as weight loss, intestinal bleeding, or diarrhea. Histological examination of small intestine and colon tissue did not reveal any signs of intestinal epithelial damage or epithelial hyperplasia in 8–10-week-old *CARD14*(E138A)$^{IEC}$ mice (Fig. 1B). In agreement, intestinal barrier permeability was comparable in 8-week-old *CARD14*(E138A)$^{IEC}$ and littermate WT control mice (Fig. 1C).

We have previously shown that keratinocyte-specific expression of *CARD14*(E138A) in mice leads to the development of psoriasis-like skin pathology, which is characterized by epidermal thickening and increased inflammatory cell infiltration, and cytokine expression (Van Nuffel et al, 2020). Similarly, flow cytometry analysis of the intestinal lamina propria of *CARD14*(E138A)$^{IEC}$ mice revealed increased infiltration of neutrophils, eosinophils, and dendritic cells (Fig. 1D). In addition, expression of several inflammatory mediators, such as *Tnf*, *Ccl20*, *Nos2* and *Cxcl2*, was increased in

small intestine IEC of *CARD14*(E138A)$^{IEC}$ mice compared to WT littermates (Fig. 1E). Altogether, our data demonstrate that *CARD14*(E138A) expression in IEC results in low-grade intestinal inflammation.

## Intestinal transit is drastically reduced in *CARD14*(E138A)$^{IEC}$ mice

Surprisingly, 31% of *CARD14*(E138A)$^{IEC}$ mice developed rectal prolapse at the age of 5–7 weeks (Fig. 2A). As rectal prolapse is often associated with pathological conditions that increase intra-abdominal pressure, such as constipation (Goldstein and Maxwell, 2011), we reasoned that *CARD14*(E138A) expression in IEC might influence intestinal motility. Therefore, we next analyzed gastro-intestinal transit (GIT) time in *CARD14*(E138A)$^{IEC}$ mice by gavaging carmine red, a non-absorbable red dye, and subsequent monitoring for the first appearance of colored feces. Gastrointestinal transit time was delayed by nearly 40% in *CARD14*(E138A)$^{IEC}$ mice compared to littermate controls (Fig. 2B). Although intestinal motility can be affected by sex hormones (Oh et al, 2013), with the prevalence of constipation being higher in women (Suares and Ford, 2011), gastrointestinal transit time was similarly delayed in *CARD14*(E138A)$^{IEC}$ male and female mice (Fig. 2B). Therefore, both sexes were used in subsequent experiments. To investigate whether delayed gastric emptying might explain the increased gastrointestinal transit time in *CARD14*(E138A)$^{IEC}$ mice, we gavaged mice with 70 kDa FITC-dextran, which does not cross the epithelial barrier, and measured gastric emptying as the percentage of total fluorescent signal remaining in the stomach after 5 min. However, no differences in gastric emptying were detected between both genotypes (Fig. 2C). Collectively, these data indicate that *CARD14*(E138A)$^{IEC}$ mice show reduced intestinal motility, resulting in a longer intestinal transit time and a constipation-like phenotype.

Delayed gastrointestinal transit in post-operative ileus patients was previously linked to increased intestinal neutrophil infiltration following surgery (Farro et al, 2017). To analyze a possible causal role of increased neutrophil infiltration in delayed intestinal motility in *CARD14*(E138A)$^{IEC}$ mice, we tested the effect of neutrophil depletion. In this context, we first generated inducible *CARD14*(E138A)$^{IEC}$ mice (referred to as *iCARD14*(E138A)$^{IEC}$), where *CARD14*(E138A) expression in IEC can be induced by tamoxifen administration, allowing better temporal control of *CARD14*(E138A)-induced inflammation. Moreover, the use of *iCARD14*(E138A)$^{IEC}$ mice also excludes potential compensatory effects that could mask a specific phenotype of *CARD14*(E138A) transgene expression. Gastrointestinal transit of untreated *iCARD14*(E138A)$^{IEC}$ mice was comparable to littermate controls, and induction of *CARD14*(E138A) expression by a single oral administration of tamoxifen resulted in delayed intestinal motility already after five days (Fig. 2D). To deplete neutrophils, mice were intraperitoneally injected on days 1, 2, and 3 (with day 1 the time of oral tamoxifen administration) with anti-Gr-1 monoclonal antibody (clone RB6-8C5), which reacts predominantly with Ly6G on neutrophils and weakly with Ly6C on neutrophils, dendritic cells, and specific subsets of lymphocytes and monocytes (Daley et al, 2008). Efficient neutrophil depletion in total blood and lamina propria of the small intestine was confirmed by flow cytometry (Fig. 2E). Most importantly, anti-

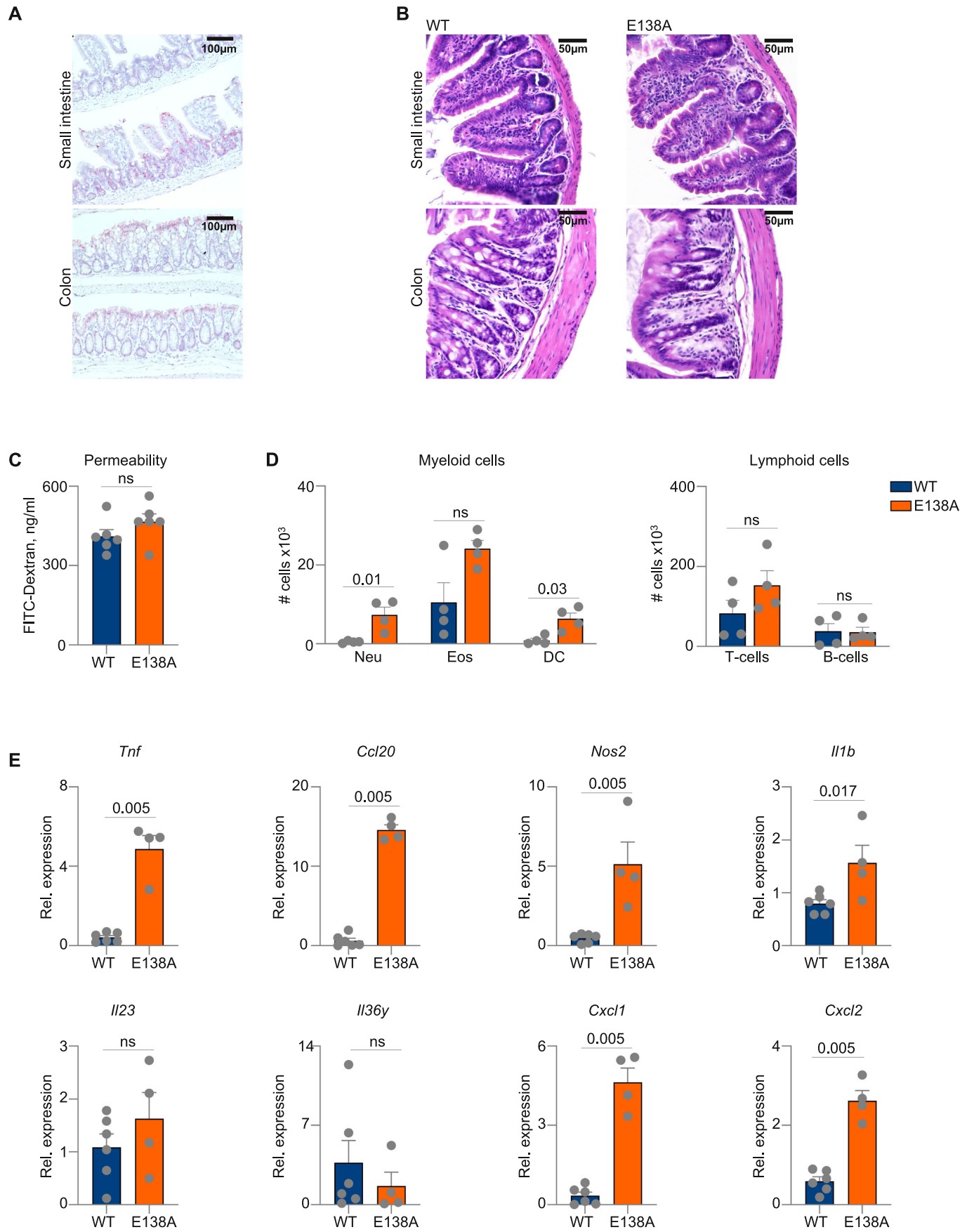

**Figure 1.  *CARD14*(E138A) expression in IEC induces low-grade intestinal inflammation.**

(A) RNAscope showing *Card14* mRNA localization in the small intestine and colon of WT mice. Scale bar: 100 μm. (B) H&E staining of the small intestine and colon. Scale bar: 50 μm. (C) Serum FITC-dextran level 2 h after oral gavage. (D) Number of immune cell populations in the lamina propria of the small intestine, analyzed by flow cytometry. (E) Relative mRNA expression in IEC lysates from the small intestine. All data are presented as mean ± SEM, each symbol represents one mouse. Data are representative of two independent experiments. Statistical differences were analyzed using an unpaired *t* test (C), a Hierarchical Generalized Linear Model (D), or a Mann–Whitney *U* test with multiple comparisons (E). ns not significant. Source data are available online for this figure.

Gr-1 treatment had no effect on *CARD14*(E138A)-induced slow gastrointestinal transit (Fig. 2D), excluding a causal role of neutrophilic inflammation.

## *CARD14*(E138A)[IEC]-induced delay in intestinal transit is dependent on MALT1 scaffold function

CARD14 activation induces the formation of a so-called CBM complex, consisting of CARD14, BCL10, and paracaspase MALT1, which enables MALT1 to act as a molecular scaffold that interacts with NF-κB and MAPK signaling proteins. In addition, MALT1 exerts a unique proteolytic activity that further contributes to inflammatory signaling by cleaving specific substrates that are involved in CBM auto-regulation, regulation of signaling and adhesion, or transcription and mRNA stability/translation (Afonina et al, 2015; Afonina et al, 2016; Howes et al, 2016; Moud et al, 2024). We, as well as others, have previously shown that *CARD14*-induced gene expression in keratinocytes is dependent on the scaffold and proteolytic activity of MALT1 (Afonina et al, 2016; Howes et al, 2016). Furthermore, *CARD14*(E138A)-induced psoriatic dermatitis in mice is prevented by the absence of MALT1 or upon pharmacological inhibition of MALT1 protease activity (Van Nuffel et al, 2020; Zhang et al, 2021b). To investigate the role of MALT1 in *CARD14*(E138A)-induced inflammation and motility changes, *CARD14*(E138A)[IEC] mice were crossed with mice containing a floxed (fl) *Malt1* allele (Demeyer et al, 2019) to obtain mice that lack MALT1 expression in IEC. Importantly, the absence of MALT1 completely abolished the ability of *CARD14*(E138A) to delay gastrointestinal transit (Fig. 3A). Similarly, increased *Tnf* and *Cxcl2* expression, as well as neutrophil infiltration, were no longer observed in *CARD14*(E138A)[IEC] mice lacking MALT1 in IEC (Fig. 3B,C).

We next tested whether pharmacological inhibition of MALT1 proteolytic activity can also normalize intestinal transit in *CARD14*(E138A)[IEC] mice. To this end, we again took advantage of *iCARD14*(E138A)[IEC] mice, in which *CARD14*(E138A) expression in IEC was induced by oral administration of tamoxifen on 3 consecutive days, which was accompanied by daily intraperitoneal injection of the small compound MALT1 inhibitor MLT-827 or vehicle for 7 days (day 1 = first day of tamoxifen). Importantly, *CARD14*(E138A) expression in IEC-induced proteolytic activity of MALT1, as indicated by the cleavage of the MALT1 substrate CYLD, which was efficiently blocked by MALT1 inhibitor treatment (Fig. 3D). However, MALT1 inhibition had no effect on decreased intestinal motility in *iCARD14*(E138A)[IEC] mice (Fig. 3D), nor on the increased expression of *Tnf* and *Cxcl2* (Fig. 3E). Taken together, our data show that *CARD14*(E138A)-induced slow intestinal transit and inflammatory gene expression in IEC is dependent on the scaffold function of MALT1, but independent of its catalytic protease activity.

## Enteric neuronal survival and functionality is unaffected in *CARD14*(E138A)[IEC] mice

Gut inflammation can lead to enteric neuronal dysfunction and cell death (Matheis et al, 2020; Ye et al, 2020), thus affecting motility. In addition to infiltrating immune cells, resident gut immune cells have previously also been shown to regulate gut motility and intestinal ion secretion by supporting enteric neurons (De Schepper et al, 2018; Matheis et al, 2020). To assess submucosal neuron function in *CARD14*(E138A)[IEC] mice, we measured anion secretion across intestinal epithelium in ileal preparations upon stimulation of submucosal neurons with the voltage-gated sodium channel agonist veratridine. Neuron-evoked anion secretion, calculated form the area under the curve, was significantly lower in *CARD14*(E138A)[IEC] mice compared to control mice (Fig. 4A). Transepithelial electrical resistance (TEER) measurement was also performed and showed no difference between the two groups (Fig. 4B), further confirming earlier results showing the integrity of the intestinal barrier in vivo by FITC-dextran permeability assay (Fig. 1C). Staining of the pan-neuronal marker HuC/D did not reveal any changes in the number of ganglia or density of HuC/D+ enteric neurons per ganglia in the submucosal plexus of small intestine from *CARD14*(E138A)[IEC] and control mice (Fig. 4C,D), suggesting that survival of enteric neurons is not affected in *CARD14*(E138A)[IEC] mice.

While submucosal plexus neurons regulate IEC function and secretion, smooth muscle contraction facilitating intestinal motility is primarily controlled by myenteric plexus (Avetisyan et al, 2015). Therefore, we next compared (neuron-evoked) smooth muscle contractility of intestinal myenteric plexus-muscle strips in organ bath experiments (Rychter et al, 2014). First, we applied electric field stimulation to ileal muscle strips, creating a frequency-response curve that induces neurotransmitter release from neurons, triggering muscle contractions. Electric field stimulation-induced contractions were similar in ileum of *CARD14*(E138A)[IEC] mice and WT control mice (Fig. 4E). Also, direct muscle stimulation by exposing the tissue to the muscarinic receptor agonist carbachol did not reveal any difference between *CARD14*(E138A)[IEC] and control mice (Fig. 4F). Together, these data demonstrate that the survival and cell-intrinsic functionality of enteric neurons is unaffected in ileum of *CARD14*(E138A)[IEC] mice, and indicate that reduced intestinal motility in *CARD14*(E138A)[IEC] mice may result from IEC-intrinsic changes that affect epithelial–neuronal communication in the intestine, rather than from enteric neuronal loss or dysfunction.

## Transcriptome analysis of IEC from *CARD14*(E138A)[IEC] mice reveals functional changes in intestinal secretory cells

To further examine the molecular changes driven by pathogenic CARD14 signaling that are responsible for slow gastrointestinal

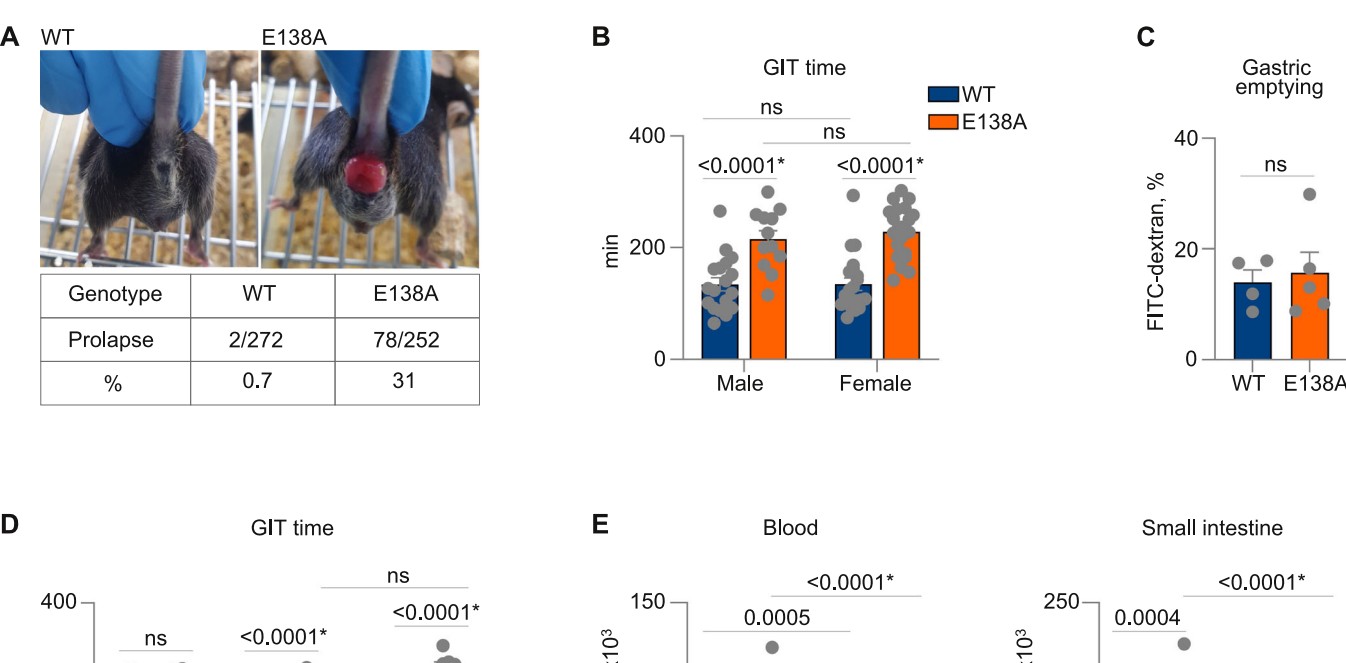

**Figure 2. CARD14(E138A)^IEC mice have reduced intestinal motility.**

(A) Incidence of rectal prolapse in control (WT) and CARD14(E138A)^IEC mice, with representative images. (B) Total GIT time in female and male WT and CARD14(E138A)^IEC mice. Data pooled from six different experiments. (C) Gastric emptying assessed 5 min after gavage with 70 kDa FITC-dextran. (D, E) GIT time in mice, treated with control or anti-Gr-1 antibody (days 1 and 3). CARD14 expression was induced by a tamoxifen injection on day 1. GIT time was measured on day 5. (D) Number of neutrophils in the blood and lamina propria of the small intestine, analyzed by flow cytometry (E). All data are presented as mean ± SEM; each symbol represents one mouse. Data are representative of two independent experiments. Statistical differences were analyzed using an unpaired t test (C) or two-way ANOVA with Sidak's correction for multiple comparisons (B, D, E, left panel) or Tukey's test (E, right panel). Data in (E, right panel) were log-transformed before statistical analysis. ns not significant, *P = 9.40726E-10 (B, male, female), *P = 8.67408E-009 (D, tamoxifen-Gr1, tamoxifen+Gr-1), *P = 1.76973E-005 (E, blood), *P = 2.27244E-007 (E, small intestine). Source data are available online for this figure.

transit, we performed RNA sequencing (RNAseq) on polyA-enriched RNA of IEC isolated from the small intestine and colon of CARD14(E138A)^IEC and WT control mice. In the small intestine, there were 1289 differentially expressed genes (DEG), of which 217 were upregulated at least twofold and 119 were downregulated at least twofold. Of these, 58 genes were upregulated fivefold, while 19 genes were downregulated fivefold (Figs. EV2A and 5A). There were 1627 differentially expressed genes in colon IEC of CARD14(E138A)-expressing mice compared to controls, of which 290 were upregulated at least twofold and 202 were downregulated at least twofold. Notably, 62 genes were upregulated by fivefold, while 44 genes were downregulated fivefold (Figs. EV2B and 5B). A comprehensive overview of all the upregulated and downregulated genes in the small intestine and colon is listed in the Tables EV1 and EV2, respectively. Genes associated with an NF-κB-regulated inflammatory signature (e.g., Tnf, Cxcl1, Ccl20, Il1b) were strongly upregulated in CARD14(E138A)-expressing IEC of both small intestine and colon (Fig. 5A–C), which is in agreement with the

known role of CARD14 in NF-κB signaling in keratinocytes and in line with the intestinal inflammation described above. Unexpectedly, RNAseq analysis also revealed a strong downregulation of several antimicrobial peptides (AMPs) (e.g., Lyz1, Defa26, Defa35) (Fig. 5A,C,D), in the small intestine of CARD14(E138A)^IEC mice, which was further confirmed by qPCR (Fig. 5E). AMPs are produced by Paneth cells, which are specialized secretory cells in the small intestine that play key roles in maintaining gut homeostasis and mediating host-microbiome cross-talk (Wallaeys et al, 2023). Significant changes were also detected in the expression of gut hormones (downregulated Gcg and Nts in the small intestine, Fig. 5A; upregulated Pyy and Cck and downregulated Gcg, Sst, and Gip in the colon; Fig. 5B), which are produced by enteroendocrine cells (EEC) in the intestine, and genes related to bile acid metabolism (decreased Abcg5, ApoB, Fabp6, and increased Fgf15 in the colon; Fig. 5B). Both were also validated by qPCR (Fig. EV2C,D). Differentially expressed genes in CARD14(E138A)^IEC and WT control mice were further analyzed

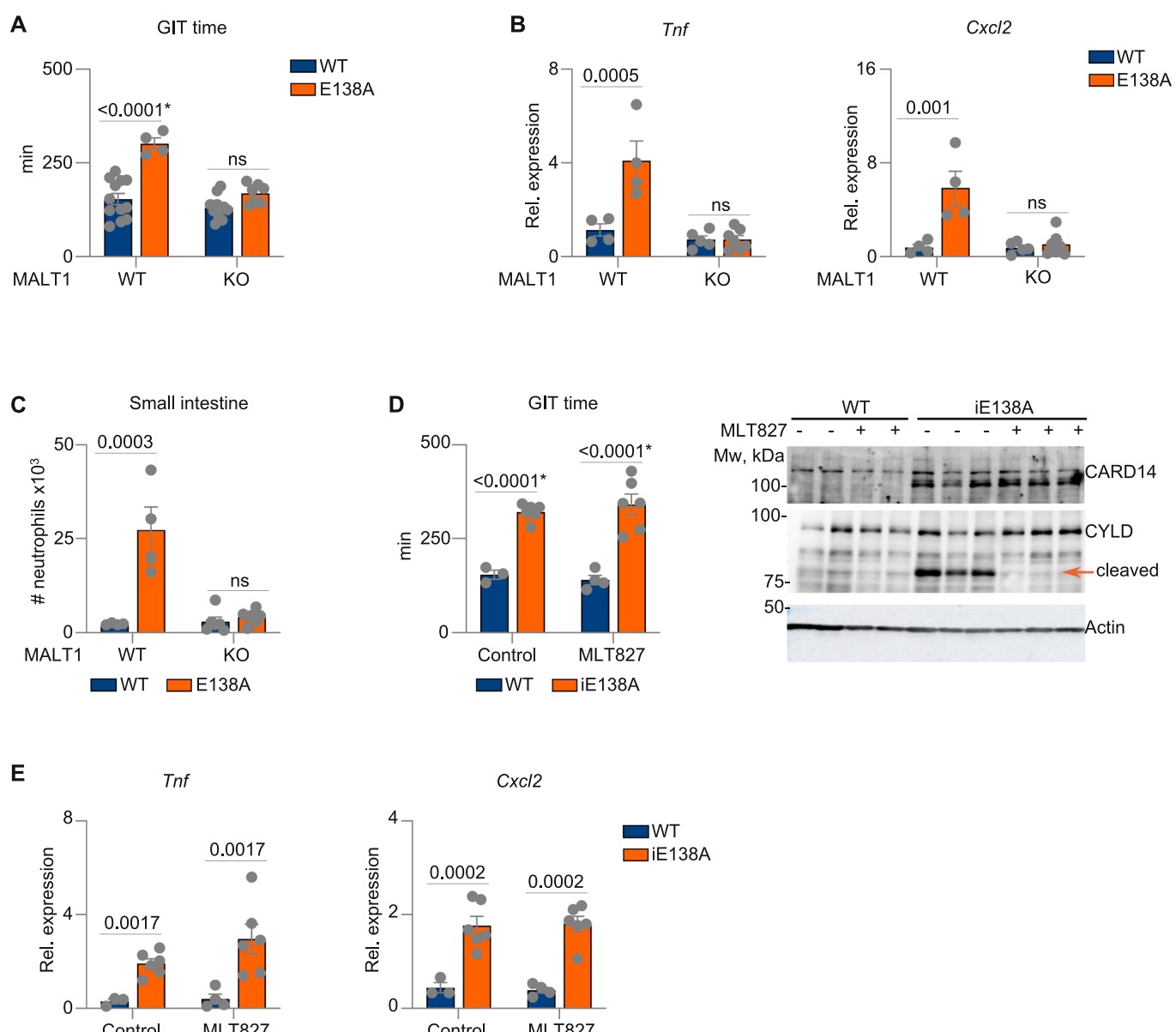

**Figure 3. CARD14(E138A)-induced inflammation and slow intestinal transit are MALT1-dependent.**

(A–C) GIT time (A), relative mRNA expression of *Tnf* and *Cxcl2* in IEC lysates from the small intestine (B), as well as the total number of neutrophils in the lamina propria of the small intestine (C) were measured in control (WT) and *CARD14*(E138A)[IEC] mice that were either MALT1 WT or MALT1-deficient (KO). (D, E) Effect of MALT1 inhibition. WT and *iCARD14*(E138A)[IEC] mice were treated daily i.p. with vehicle or 30 mg/kg of MALT1 inhibitor, MLT-827, for 7 days. *CARD14*(E138A) expression was induced by three tamoxifen injections on days 1–3. GIT time was measured on day 7. Cleavage of a MALT1 substrate CYLD (D) was assessed by immunoblotting of IEC lysates from the small intestine (cleaved = cleavage fragment of CYLD). Each lane represents one mouse. Actin was used as a loading control. (E) Relative mRNA expression of *Tnf* and *Cxcl2* in IEC lysates from the small intestine was analyzed by qPCR. Data are presented as mean ± SEM; each symbol represents one mouse. Data are representative of two independent experiments. Statistical differences were analyzed with two-way ANOVA with Tukey's test (A, C, D) or a log-linear regression model (B, E). ns not significant, *P = 1.45557E-06 (A), *P = 2.60256E-007 (D, control, MLT-827). Source data are available online for this figure.

using Ingenuity Pathway Analysis, which confirmed the activation of canonical bile acid-related pathways, such as LXR/RXR and FXR/RXR signaling (Fig. EV2E). Moreover, gene ontology pathway analysis in the colon confirmed the activation of inflammatory signaling and also indicated the downregulation of genes associated with lipid metabolism and transport, which may be functionally linked to alterations in bile acid metabolism (Fig. EV2F).

## CARD14(E138A) signaling in IEC induces Paneth cell dysfunction

The gut microbiota significantly influences gut health and exhibits a bidirectional relationship with other processes such as gut hormone production and bile acid metabolism (Chao et al, 2025). Because of this, the central role of the gut microbiota and the drastic reduction of AMPs in the small intestine of *CARD14*(E138A)[IEC] mice, we

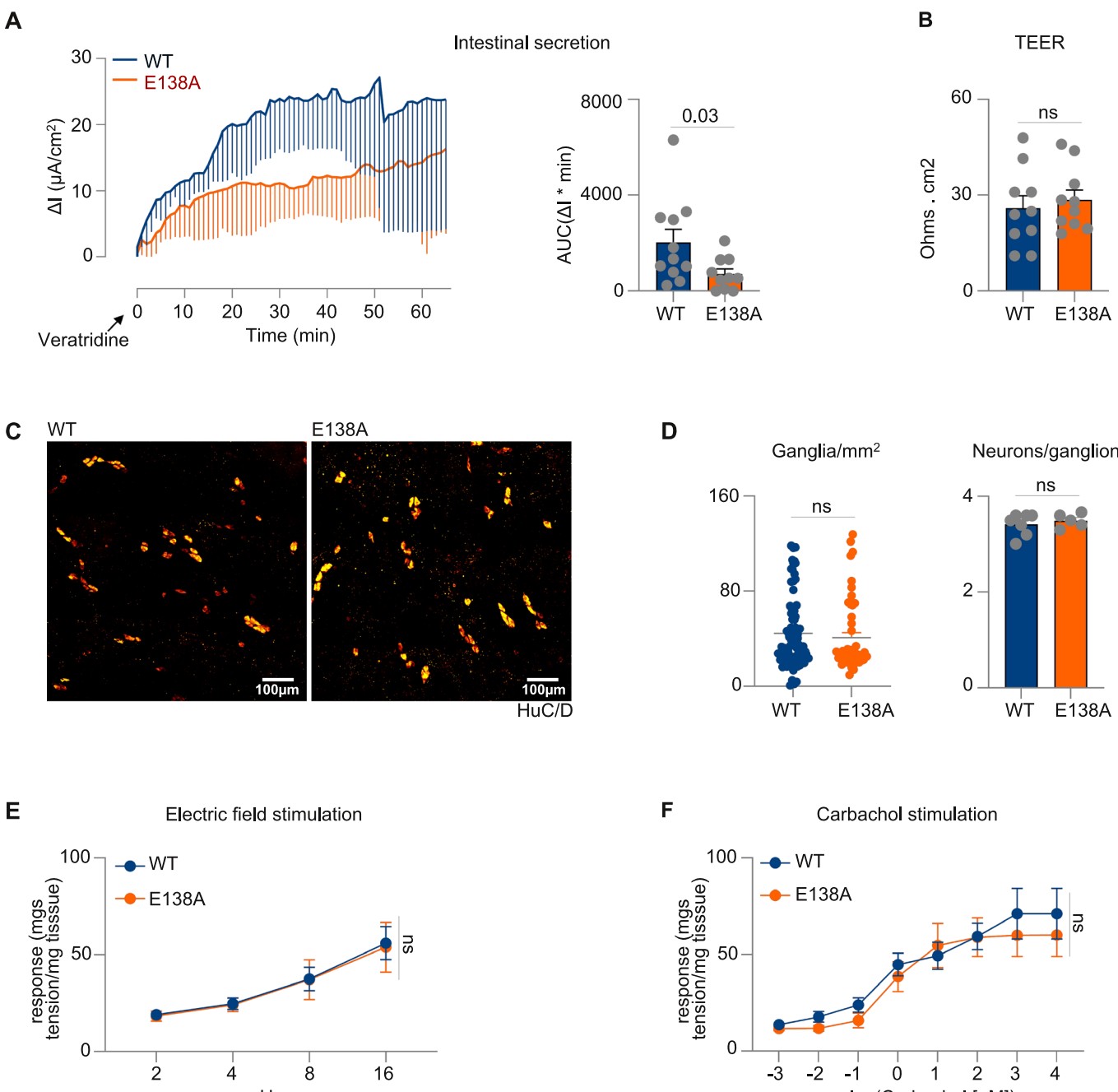

**Figure 4. Enteric neuronal survival and functionality is unaffected in *CARD14*(E138A)^IEC mice.**

(A) Veratridine-evoked (30 µM) short-circuit current (Isc) responses over 65 min in ileal lamina propria-submucosa preparations from control (WT) ($n = 13$) and *CARD14*(E138A)^IEC mice ($n = 10$). Right panel: area under the curve (AUC) quantification. (B) Transepithelial electrical resistance (TEER) across epithelial cell layers in WT and *CARD14*(E138A)^IEC mice, measured in Ω.cm². (C) Representative whole-mount images of submucosal plexus from WT and *CARD14*(E138A)^IEC mice stained with anti-HuC/D, a pan-neuronal marker. Scale bar: 100 µm. (D) Absolute quantification of ganglia/mm² (each symbol represents one ganglion) and total numbers of HuC/D+ neurons in different ganglia (each symbol represents one mouse) of the submucosal plexus from WT ($n = 7$) and *CARD14*(E138A)^IEC ($n = 5$) mice. At least 100 ganglia were counted per mouse. (E, F) Contractile responses of ileal muscle strips from WT ($n = 6$) and *CARD14*(E138A)^IEC ($n = 3$) mice. (E) Frequency-response curve to electric field stimulation (Hz), ranging from 2 to 16 Hz. (F) Concentration-response curve to the cholinergic agonist carbachol. Data are from one experiment and presented as mean ± SEM. Statistical differences were analyzed using the Mann–Whitney *U* test (A), the unpaired *t* test (B, D), or as repeated measurements using the residual maximum likelihood (REML) (E, F). ns not significant. Source data are available online for this figure.

**A**

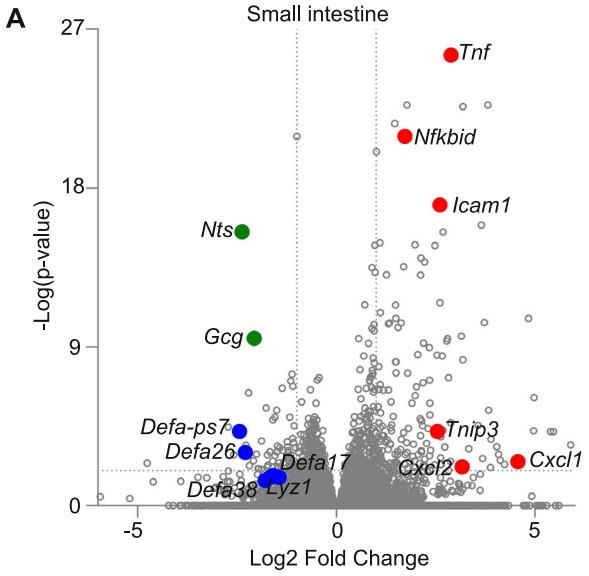

Small intestine

**B**

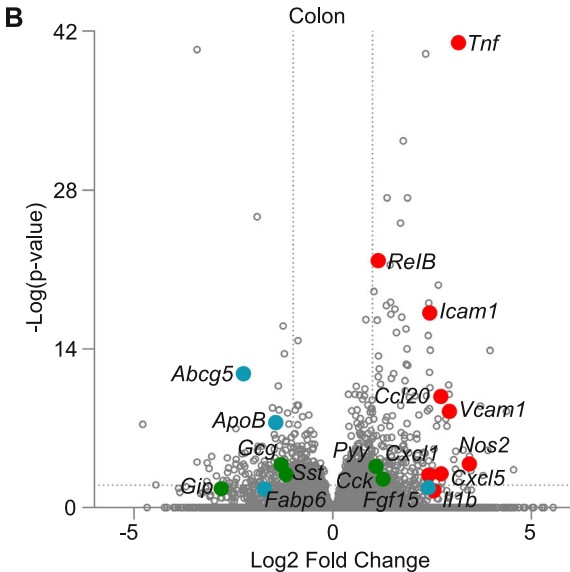

Colon

**C**

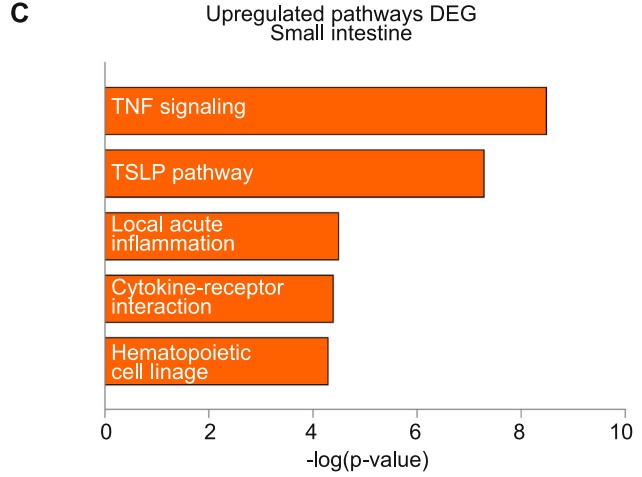

Upregulated pathways DEG
Small intestine

Downregulated pathways DEG
Small intestine

**D**

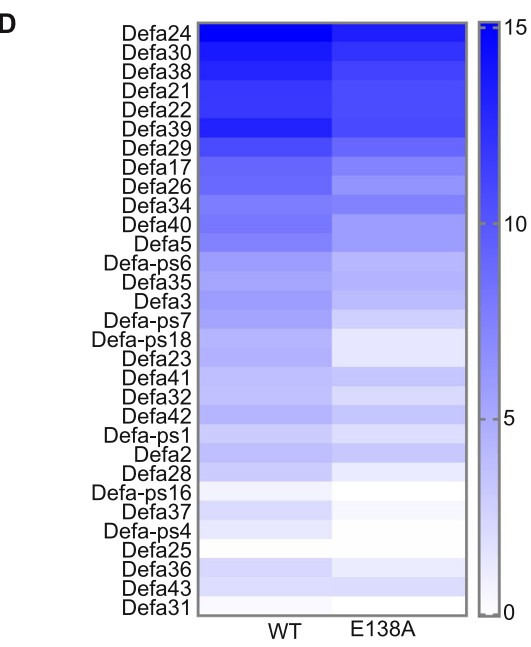

**E**

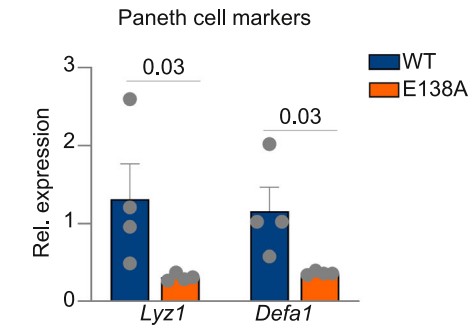

Paneth cell markers

◀ **Figure 5.  Transcriptional changes in IEC of *CARD14*(E138A)^IEC mice.**

(A, B) Volcano plots showing differentially expressed genes (DEGs) in IEC of small intestine (A) and colon (B) from control (WT) and *CARD14*(E138A)^IEC mice (n = 4). (C) Gene ontology enrichment analysis of biological processes for upregulated (top) and downregulated (bottom) DEGs in small intestine IEC isolated from WT and *CARD14*(E138A)^IEC mice. (D) Heatmap of normalized Log$_2$ (counts) of DEGs linked to α-defensins in IEC of small intestine. (E) Relative mRNA expression of *Lyz1* and *Defa1* in IEC lysates of the small intestine. RNA-seq data are derived from one experiment, whereas qPCR data are representative of three independent experiments. Each symbol represents one mouse. Data are presented as mean ± SEM and analyzed using a Wald test with Benjamini–Hochberg multiple correction (A, B, D), a Fisher's exact test (C), and a Mann–Whitney *U* test with multiple comparisons (E). Source data are available online for this figure.

further focused on Paneth cells. In agreement with the observed decrease in *Lyz1* mRNA expression (Fig. 5E), immunostaining of intestinal tissue for lysozyme P (encoded by *Lyz1*), an AMP that is predominantly stored in Paneth cell granules, revealed a strong depletion of lysozyme P in the crypts of *CARD14*(E138A)-expressing mice when compared to WT control mice (Fig. 6A,B). Furthermore, lysozyme activity in the ileum was significantly reduced, as measured by a hydrolysis assay utilizing fluorescein-labeled *Micrococcus lysodeikticus* cell walls (Fig. 6C). However, this analysis does not allow us to verify whether decreased AMP expression is due to a dysfunction of Paneth cells or the absence of Paneth cells themselves. More specifically, perturbance of intestinal epithelial signaling might result in dysregulated differentiation of IEC into different secretory or absorptive subtypes, including Paneth cells (Pinto et al, 2003; Kuhnert et al, 2004; van Dop et al, 2009). To visualize the cellular architecture of intestinal epithelium in *CARD14*(E138A)^IEC mice, we performed transmission electron microscopy, showing that enterocyte morphology and brush border are normal in *CARD14*(E138A)^IEC mice (Fig. 6D). Furthermore, Paneth cells containing large vacuoles interspersed with smaller cuboidal stem cells could be observed in the ileal crypts from both *CARD14*(E138A)^IEC and WT control mice (Fig. 6D). Interestingly, the number of Paneth cells in intestinal crypts was only slightly lower in *CARD14*(E138A)^IEC mice, as compared to control mice (Fig. 6E). Also the number of Goblet cells was comparable (Fig. 6E). These data exclude Paneth cell loss as an underlying mechanism for the observed reduction in AMP in *CARD14*(E138A)^IEC mice and rather points to a more subtle regulation of Paneth cell function. To investigate whether Paneth cell dysfunction is mediated by Paneth cell-intrinsic CARD14 signaling, we crossed floxed *CARD14*(E138A) mutant mice with mice that express Cre recombinase under the control of the Paneth cell-specific *Defa6* promotor (Adolph et al, 2013), leading to mice that specifically express *CARD14*(E138A) in Paneth cells (further referred to as *CARD14*(E138A)^PC mice) (Fig. 6F). In contrast to *CARD14*(E138A)^IEC mice, mice expressing *CARD14*(E138A) only in Paneth cells did not show any differences in gastrointestinal transit time, when compared to their littermate controls (Fig. 6G). Moreover, expression of Paneth cell markers *Lyz1* and *Defa1* (Fig. 6H), as well as inflammatory genes *Tnf* and *Ccl20* (Fig. 6I) were unchanged in *CARD14*(E138A)^PC mice. Collectively, these data suggest that Paneth cell-intrinsic hyperactive *CARD14*(E138A) signaling is insufficient to induce Paneth cell dysfunction and intestinal dysmotility.

Inflammatory cytokines such as TNF, whose expression is upregulated in the IEC of *CARD14*(E138A)-expressing mice (Figs. 1E and 5A,B), were previously shown to induce Paneth cell dysfunction (Vereecke et al, 2014; Van Hauwermeiren 2015). Interestingly, increased TNF expression has also been associated with reduced fecal output and water content in aged mice, which

was reversed by administration of a TNF antagonist (Patel et al, 2017). To investigate whether TNF is involved in Paneth cell dysfunction and intestinal dysmotility in *CARD14*(E138A)^IEC mice, we tested the effect of a TNF-neutralizing monoclonal antibody (clone XT3.11) on intestinal motility in i*CARD14*(E138A)^IEC mice. However, anti-TNF treatment was not able to prevent the *CARD14*(E138A)-induced delay in intestinal transit (Fig. 6J). As an alternative approach, we also analyzed intestinal transit time in TNF^emARE mice, which overexpress TNF (Fig. EV3) and spontaneously develop intestinal inflammation (Thiran et al, 2023). Nevertheless, TNF^emARE mice and WT control mice showed similar intestinal motility (Fig. 6K). Together, these data indicate that TNF is not a central driver of delayed intestinal motility in *CARD14*(E138A)^IEC mice, and that other mechanisms likely contribute to Paneth cell dysfunction and impaired motility.

## Paneth cell dysfunction in *CARD14*(E138A)^IEC mice is associated with microbial dysbiosis and increased susceptibility to enteric bacterial infection

Defects in AMP production in *CARD14*(E138A)^IEC mice might lead to alterations in gut microbiome composition. Therefore, we analyzed the fecal microbiome of *CARD14*(E138A)^IEC and control mice by 16S rRNA gene Illumina sequencing. The total number of microbial species present, measured by Chao1 richness index, was similar in *CARD14*(E138A)^IEC and control mice (Fig. 7A). However, a significant reduction in microbial diversity, which reflects both richness and proportional abundance of microbial species (measured by inverted Simpson index), was observed in *CARD14*(E138A)^IEC mice (Fig. 7A). Intestinal microbiota has been shown to affect gut motility through the release of end-products of fermentation, activation of immune cells, and secretion of neuroendocrine substances (Barbara et al, 2005). To further analyze a potential causal link between microbial dysbiosis in *CARD14*(E138A)^IEC mice and reduced gut motility, we depleted commensal microbiota in i*CARD14*(E138A)^IEC mice by treatment with a broad-spectrum antibiotic cocktail for 3 weeks, after which *CARD14*(E138A) expression was induced by oral administration of tamoxifen. Microbiome depletion was confirmed by the absence of microbial colonies on brain heart infusion agar plates. Antibiotic treatment on its own already increased the gastrointestinal transit time of control wild-type mice. Most importantly, however, inducible *CARD14*(E138A) expression did not further increase gastrointestinal transit time in antibiotic-treated mice (Fig. 7B), suggesting that changes in microbial composition in *CARD14*(E138A)^IEC mice could contribute to decreased intestinal motility. To further validate the role of the microbiome, we rederived *CARD14*(E138A)^IEC and control WT mice under germ-free conditions in the germ-free and gnotobiotic mouse core facility of our center. In agreement with the results after antibiotic

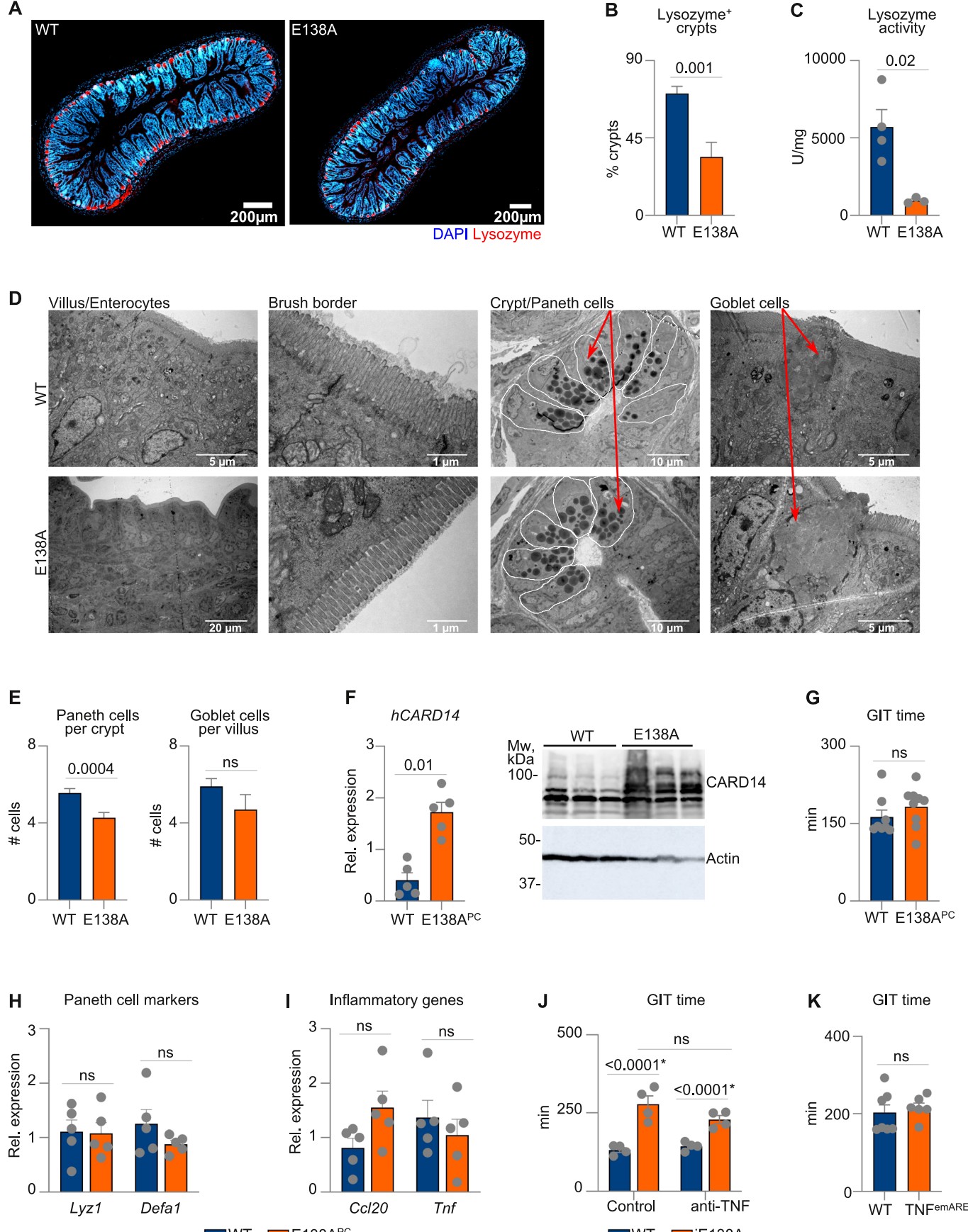

**Figure 6. CARD14(E138A) signaling in IEC induces Paneth cell dysfunction.**

(A) Representative images of immunofluorescent staining for DAPI and lysozyme in small intestinal sections from control (WT) and *CARD14*(E138A)$^{IEC}$ mice. Scale bars: 200 μm. (B) Percentage of lysozyme$^+$ crypts in the small intestine of WT ($n = 3$) and *CARD14*(E138A)$^{IEC}$ ($n = 3$) mice. (C) Lysozyme activity in IEC measured using fluorescent-labeled peptidoglycan from *Micrococcus lysodeikticus* in WT and *CARD14*(E138A)$^{IEC}$ mice. (D, E) Transmission electron microscopy of ileum. (D) Representative images of the enterocyte morphology, microvilli structure, and ultrastructure of Paneth and goblet cells. (E) Quantification of Paneth cells per crypt and goblet cells per villus in WT ($n = 4$) and *CARD14*(E138A)$^{IEC}$ mice ($n = 2$). Paneth cells were counted from 5 to 12 crypts per mouse, and goblet cells from 5 to 6 villi. (F) Relative mRNA and protein levels of human CARD14 in IEC lysates of the small intestine from WT and *CARD14*(E138A)$^{PC}$ mice. (G) Total GIT time of WT and *CARD14*(E138A)$^{PC}$ mice. (H, I) Relative mRNA expression of Paneth cell (H) and inflammatory response genes (I) in IEC lysates of small intestine from WT and CARD14(E138A)$^{PC}$ mice. (J) Total GIT time of WT and *iCARD14*(E138A)$^{IEC}$ mice treated with control or anti-TNF antibody (days 1 and 4), with tamoxifen induction on day 1. (K) Total GIT time in control (WT) and TNF$^{emARE}$ mice. Data are presented as mean ± SEM; each symbol represents one mouse. Data are representative of one experiment (A–E, J), three independent experiments (F–I) or two independent experiments (K). Statistical differences were analyzed using unpaired *t* test (B, C, E, G, K), a Mann–Whitney *U* test with multiple comparisons (F, H, I) or two-way ANOVA with Sidak's multiple comparison test (J). ns not significant, *$P = 7.70133E$-006 (J, control, anti-TNF). Source data are available online for this figure.

treatment, no significant differences in gastrointestinal transit time were observed between *CARD14*(E138A)$^{IEC}$ and control WT mice in the absence of the microbiome (Fig. 7C). These data further support the role of the microbiome in *CARD14*(E138A)-induced intestinal dysmotility.

We next hypothesized that reduced AMP expression might increase the susceptibility of *CARD14*(E138A)$^{IEC}$ mice to intestinal infection. We therefore analyzed their sensitivity to infection with the natural murine enteric pathogen *Citrobacter rodentium*, which is a valuable model organism for the study of clinically significant human pathogens, such as enteropathogenic *Escherichia coli*. More specifically, *CARD14*(E138A)$^{IEC}$ and littermate control mice were orally gavaged with *C. rodentium* and monitored for 12 days, after which the experiment was terminated for ethical reasons. In line with published data (Bouladoux et al, 2017), WT C57BL/6 mice did not develop overt disease and did not show any weight loss over the course of 12 days (Fig. 7D). In contrast, *CARD14*(E138A)$^{IEC}$ mice started to lose weight one week after infection, loosing up to 20% of their initial body weight by day 12 (Fig. 7D). Similarly, fecal bacterial load remained relatively low and stable in control mice, but continued to increase in the first 7 days and was significantly higher in *CARD14*(E138A)$^{IEC}$ mice when compared to WT control mice (Fig.7E). Bacterial shedding in the feces decreased after that and was comparable between the two groups at day 12. Further microscopic examination revealed pathological changes in the colonic tissue of *CARD14*(E138A)$^{IEC}$ mice, where infection resulted in strong crypt elongation, immune cell infiltration, and Goblet cell depletion (Fig. 7F). Collectively, these data demonstrate that *CARD14*(E138A)$^{IEC}$ mice have difficulties in controlling bacterial colonization at early stages of the infection, leading to intestinal inflammation.

## Discussion

Increased *CARD14* expression has previously been shown to positively correlate with disease severity in ulcerative colitis patients (Yamamoto-Furusho et al, 2018). However, CARD14 signaling has so far been almost exclusively studied in skin, where several gain-of-function *CARD14* mutations have been associated with psoriasis. Using a novel transgenic mouse model in which the psoriasis-associated gain-of-function human CARD14(E138A) mutant protein is specifically expressed in IEC, we here show that CARD14 activation in IEC induces increased inflammatory gene expression and immune cell

infiltration into the intestinal lamina propria, further suggesting a possible link between dysregulated CARD14 signaling and colitis. However, it should be noted that intestinal inflammation in *CARD14*(E138A)$^{IEC}$ mice was rather mild and not associated with tissue damage, intestinal barrier disruption or any other clinical symptoms like weight loss or diarrhea that are typical for inflammatory bowel disease in humans. Surprisingly, we found that CARD14 signaling in *CARD14*(E138A)$^{IEC}$ mice drastically reduces intestinal transit.

The factors determining intestinal transit rate are not well understood, and changes in intestinal transit time are associated with many diseases. Slow intestinal transit has been best documented in the context of constipation, which is heavily trivialized and often ignored as a minor intestinal dysfunction. However, it can significantly impair the quality of life of affected individuals, which can be compared to that of patients with stable inflammatory bowel disease, chronic allergies, and dermatitis (Camilleri et al, 2017). Structural and molecular abnormalities in the enteric nervous system, gut hormones or microbiome in patients with constipation suggest a role for complex interactions between the immune and enteroendocrine system, as well as changes in dietary and intestinal microbial milieu (Cortesini et al, 1995; Tomita 2008; Muller et al, 2014; Farro et al, 2017; Matheis et al, 2020). Most research has focused on the neuronal regulation of smooth muscle cells, and the role of IEC often remains neglected. Also in mice, most studies related to constipation rely on a few genetic mouse models focusing on individual neurological diseases (Muller et al, 2014; Grubišić et al, 2015; Kim et al, 2019), or on opioid drug administration (Li et al, 2020; Zhang et al, 2021a; Inatomi and Honma 2021), which does not allow to study upstream events leading to neuromuscular dysfunction in the intestine. In addition, in vivo models therefore need to be developed to tackle specific questions regarding, amongst others, the role of epithelial–neuronal communication, microbiota, the immune system, and genetic predisposition in constipation. In this context, the described here *CARD14*(E138A)$^{IEC}$ mice provide the field with an interesting new model to study the specific role of IEC signaling in driving some of the above-mentioned changes in the pathology of constipation. It is important to stress that no single mouse model is suitable to study all the mechanisms involved in the pathophysiology of constipation, and that also *CARD14*(E138A)-driven intestinal dysmotility represents only a subset of mechanisms leading to constipation.

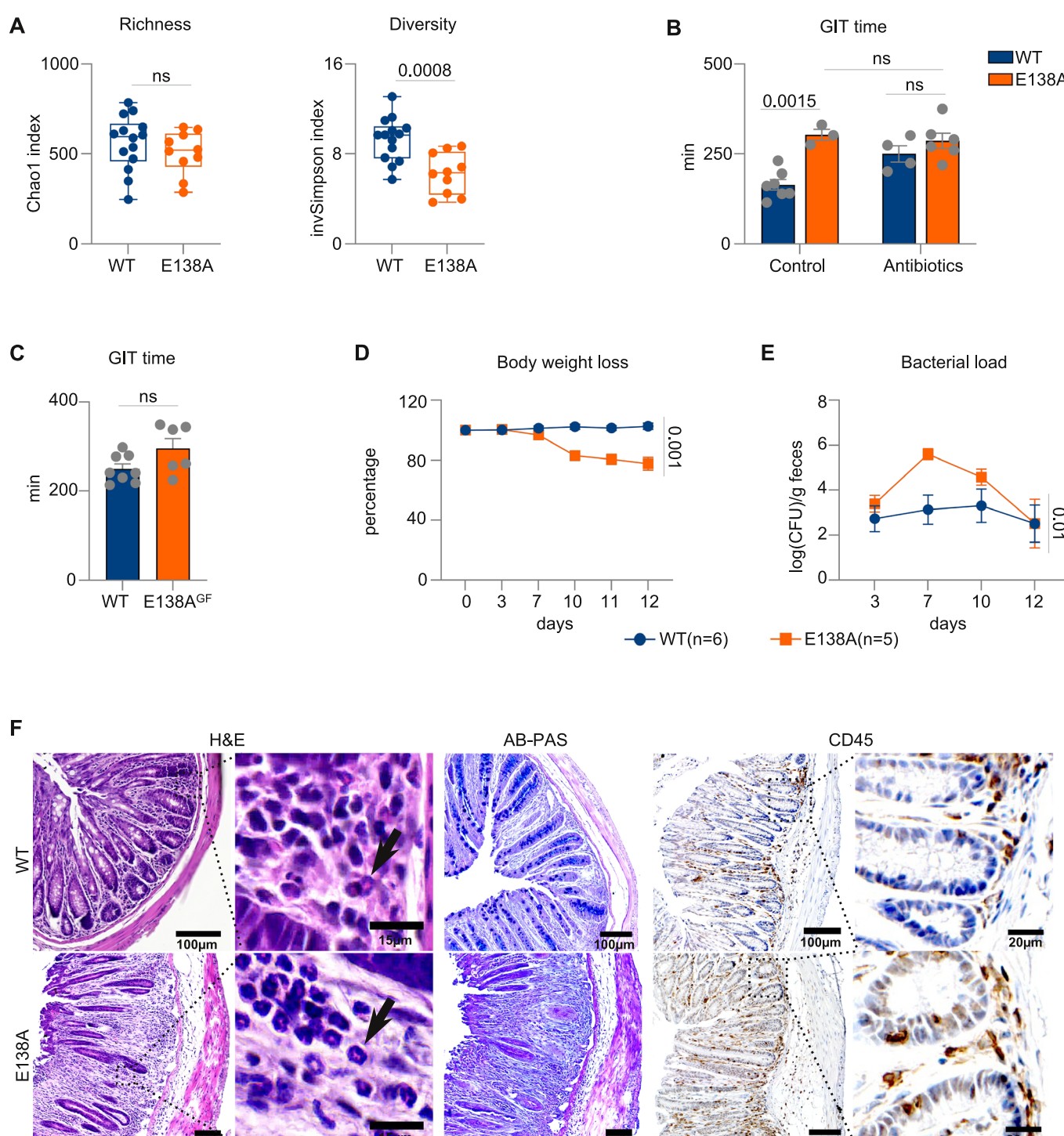

Our study further revealed a striking downregulation of Paneth cell AMPs in *CARD14*(E138A)[IEC] mice. Paneth cells and their secretion products are important for the maintenance of the intestinal stem cell niche (Sato et al, 2011; Kim et al, 2012), enteric innate immunity (Selsted et al, 1992; Salzman et al, 2003; Selsted et al, 2005), and regulation of the host microbiota (Salzman et al, 2007). In agreement, we found that Paneth cell dysfunction in *CARD14*(E138A)[IEC] mice is associated with increased susceptibility

to *C. rodentium* infection and a reduced gut microbial diversity. However, we cannot exclude that increased sensitivity also partially results from the longer transit time in *CARD14*(E138A)[IEC] mice, prolonging bacterial contact with the mucosa. Differences in stool and mucosal microbiota composition have also been observed between healthy people and patients with chronic constipation (Parthasarathy et al, 2016; Vandeputte et al, 2016). Moreover, fecal microbiota transplantation was shown to improve colonic transit

◄ **Figure 7. Microbial dysbiosis and increased susceptibility to enteric bacterial infection in CARD14(E138A)^IEC mice.**

(A) Microbiome analysis of fecal samples from control (WT) and *CARD14*(E138A)^IEC mice. Microbial richness (Chao1 index) and microbial diversity (inverted Simpson index) were calculated in the R phyloseq package. Box plots show the median (center line), 25th–75th percentiles (box), minimum to maximum values (whiskers), and all individual data points. (B) GIT time after 22-day antibiotic treatment and tamoxifen-induced *CARD14*(E138A) expression. GIT time was measured on day 6 after tamoxifen injection. (C) GIT time in germ-free WT and *CARD14*(E138A)^IEC mice. Data are combined from two independent experiments. (D, E) WT and *CARD14*(E138A)^IEC mice were orally gavaged with $5 \times 10^9$ CFU of *Citrobacter rodentium*. Percentage of body weight loss (D) and bacterial load in fecal samples (E) were monitored, as indicated. (F) Representative images of H&E, AB-PAS, and CD45 staining of colon sections (scale bars are indicated on the images). Arrows indicate granulocytes. Data are presented as mean ± SEM; each symbol represents one mouse. Data are representative of one experiment (A, D–F) or two independent experiments (C). Statistical analysis was performed using unpaired *t* test (A, C), two-way ANOVA with Tukey's multiple comparison tests (B), or as repeated measurements using the residual maximum likelihood (REML) (D, E). ns not significant. Source data are available online for this figure.

and defecation frequency of patients affected by constipation (Reigstad et al, 2015; Zhao and Yu 2016; Tian et al, 2017; Ge et al, 2017b; Zhang et al, 2018; Ding et al, 2018). Together, these data suggest a possible link between Paneth cell dysfunction, intestinal dysbiosis, and delayed intestinal transit in our mouse model. This is further supported by the fact that increased CARD14 signaling no longer increases intestinal transit time in mice treated with antibiotics or in germ-free mice. In agreement with previous reports demonstrating a key regulatory role of the microbiome in intestinal motility, both antibiotics or germ-free conditions themselves slow transit in wild-type mice (Kashyap et al, 2013; Ge et al, 2017a; Touw et al, 2017), complicating somewhat the interpretation of our results.

The exact mechanism by which increased CARD14 signaling in IEC leads to Paneth cell dysfunction and slower intestinal transit remains unclear and can be expected to involve a complex interplay between several types of IEC, immune cells, neuronal cells, and the microbiome. Our data exclude Paneth cell-intrinsic signaling of *CARD14*(E138A), as the phenotype of *CARD14*(E138A)^IEC mice could not be mimicked by Paneth cell-specific *CARD14*(E138A)^PC mice. While TNF has been implicated in promoting Paneth cell dysfunction in other contexts (Vereecke et al, 2014; Van Hauwermeiren 2015), both TNF neutralization experiments and TNF transgenic mouse experiments seem to exclude a major role for TNF in slowing down intestinal motility in our model. Similarly, neutrophil depletion allowed us to eliminate a potential role for these inflammatory cells. Although we cannot fully exclude a causal or contributing role for inflammation in the induction of Paneth cell dysfunction and intestinal motility delay in *CARD14*(E138A)^IEC mice, the above-mentioned findings make this rather unlikely. We also demonstrated that survival and cell-intrinsic functionality of enteric neurons, as well as neuro-muscular communication, remain intact in *CARD14*(E138A)^IEC mice. Instead, our data support a more major role for impaired epithelial–neuronal communication, in which changes in microbiota can also be implicated. In this context, RNAseq of IEC from *CARD14*(E138A)^IEC mice indicated a potential role for several gut hormones (*Cck*, *Pyy*, *Gcg*) that are produced by specialized intestinal EEC in response to various stimuli, including microbial metabolites or metabolites derived from microbial fermentation, such as short-chain fatty acids (Nøhr et al, 2013; Reigstad et al, 2015; Yano et al, 2015). Gut hormones, in turn, regulate numerous functions, including gastric emptying and intestinal motility. For example, cholecystokinin (CCK) is known to mediate sensory and motor responses to intestinal distension and is believed to contribute to the symptoms of constipation, bloating, and abdominal pain, while CCK receptor antagonists have been tested

in patients with constipation (Varga et al, 2004). Peptide YY (PYY) was shown before to increase intestinal water and electrolyte absorption and exhibit antisecretory activity, which could explain the reduced neuron-evoked ion secretion in *CARD14*(E138A)^IEC mice (Bilchik et al, 1993; Cox et al, 2001). Interestingly, PYY-cell density is also increased in patients with irritable bowel syndrome with constipation (El-Salhy and Gilja, 2017). Gut hormones further act through specific receptors on enteric neurons, modulating gut motility through sensory and motor pathways (Ye and Liddle 2017). It still remains to be investigated whether EEC-intrinsic CARD14 signaling is responsible for the dysregulated expression of gut hormones or whether the observed changes are caused by changes in gut microbiota directly influencing EEC function, maturation, and hormone secretion.

Interestingly, RNAseq data of IEC isolated from *CARD14*(E138A)^IEC mice and Ingenuity Pathway Analysis also hint at changes in bile acid metabolism. After their synthesis from cholesterol in the hepatocyte, bile acids pass into the small intestine, where they act as detergents to emulsify fats, aiding in their digestion and absorption, and also function as signaling molecules that activate various nuclear and G protein-coupled receptors to regulate multiple cellular processes and functions in the intestine and other tissues (Kiriyama and Nochi, 2021). Certain bile acids are known to have a laxative effect, while other studies suggest that decreased fecal conjugated bile acids are associated with constipation (Hofmann and Hagey 2008; Hofmann et al, 2008; Rao et al 2010; Vijayvargiya et al, 2018; Li et al, 2021). The size and composition of the bile acid pool are strongly regulated by their efficient enterohepatic recirculation, metabolism, and the homeostatic feedback mechanisms connecting hepatocytes and enterocytes. Importantly, also the luminal gut microbiome is known to modulate bile acid composition via deconjugation of the primary bile acids (Kiriyama and Nochi, 2021). Changes in the intestinal microbiome of *CARD14*(E138A)^IEC mice could therefore affect epithelial–neuronal communication by altering the gut bile acid composition. Given the complexity and redundancy of gut hormone and bile acid signaling, and their bidirectional relationship with the gut microbiota, it will be very difficult to pinpoint the specific role of each in the regulation of intestinal motility in *CARD14*(E138A)^IEC mice.

Importantly, our findings suggest that gain-of-function mutations in CARD14 may not only predispose patients to psoriasis but also mild intestinal inflammation, delayed intestinal transit, and increased sensitivity to intestinal infection. Although mild intestinal inflammation and delayed motility have not yet been reported in human psoriasis patients with hyperactivating CARD14 mutations, this may have remained unnoticed or neglected. Given the

high frequency of constipation in the general human population (1/6) (Peppas et al, 2008), and the fact that *CARD14* mutations only account for a very small minority of cases (frequency of specific variants is ~0.013 or less (Jordan et al, 2012b)), demonstrating a causative role of *CARD14* hyperactivation in intestinal dysmotility in human patients will also be very challenging. Our findings also support epidemiological studies showing that inflammatory bowel disease is more prevalent in psoriasis patients than in non-psoriatic controls (Yates et al, 1982). Moreover, several studies report that psoriasis is associated with subclinical intestinal inflammation, which may set the stage for the development of intestinal disease (Ichaëlsson et al, 1997; Lundquist et al, 2025). Taken together, our data further indicate that it might be worth to increase awareness for intestinal disease symptoms among psoriasis patients and treating dermatologists.

# Methods

### Reagents and tools table

| Reagent/resource | Reference or source | Identifier or catalog number |
| --- | --- | --- |
| **Experimental models** | | |
| Rosa26LSL-CARD14-E138A | van Nuffel et al, 2020 IRC Transgenic core facility | NA |
| Villin-Cre | The Jackson Laboratory | Strain #018963 |
| Vilin-Cre-ERT2 | S Robine, Institute Curie, Paris | NA |
| Defa6-Cre | RS Blumberg, Harvard Medical School, Boston | NA |
| *Malt1*^tm1a (M. musculus) | Demeyer et al, 2019 | NA |
| TNF^emARE (M. musculus) | Thiran et al, 2023 IRC Transgenic core facility | NA |
| *CARD14*(E138A)^IEC (M. musculus) | This study | NA |
| i*CARD14*(E138A)^IEC (M. musculus) | This study | NA |
| *CARD14*(E138A)^PC (M. musculus) | This study | NA |
| *CARD14*(E138A)^IEC germ-free (M. musculus) | This study | NA |
| **Antibodies** | | |
| Rat IgG2b isotype control | BioxCell | Cat. #BE0090 |
| Anti-Gr-1 | BioxCell | Cat. #BE0075 |
| Anti-TNF | BioxCell | Cat. #BE0058 |
| Anti-CD45 | Abcam | Cat. #ab10558 |
| Anti-lysozyme | Dako | Cat. #A0099 |
| Anti-rabbit AF 568 | Abcam | Cat. #ab175471 |
| Anti-Huc C/D | V. Lennon, Mayo Clinic | N/A |

| Reagent/resource | Reference or source | Identifier or catalog number |
| --- | --- | --- |
| Cy3-conjugated donkey anti-human | Jackson ImmunoLabs | Cat. #709-165-149 |
| Anti-CARD14 | Proteintech | Cat. #10400-1-AP |
| Anti-CARD14 | Novus Biologicals | Cat. #NBP2-92873 |
| Anti-CYLD | Santa Cruz | Cat. #sc-74435 |
| Anti-actin | MP Biomedicals | Cat. #MP6472J |
| HRP conjugated anti-rabbit IgG | ThermoFisher Scientific | Cat. #31432 |
| anti-mouse IgG | ThermoFisher Scientific | Cat. #31464 |
| Anti-mouse CD16/CD32 | BD Bioscience | Cat. #553142 |
| Fixable viability dye eFluor506 | eBioscience | Cat. #65-0866-18 |
| anti-CD45-FITC | Invitrogen | Cat. #11-0451-82 |
| anti-CD11b-V450 | BD Bioscience | Cat. #560455 |
| Anti-CD64-BV711 | Biolegend | Cat. #139311 |
| Anti-CD3-PeCy5 | Tonbo Biosciences | Cat. #550031 |
| Anti-CD19-PeCy5 | eBioscience | Cat. #15-0193 |
| Anti-SiglecF-PE | BD Bioscience | Cat. #552126 |
| Anti-CD11c-PeCy7 | Biolegend | Cat. #117317 |
| Anti-Ly6C-AlexaFluor700 | BD | Cat. #561236 |
| Anti-MHCII-APCeFluor780 | eBioscience | Cat. #47-5321-82 |
| Anti-Ly6G-APC | Biolegend | Cat. #127613 |
| NP-40 alternative | Merck | Cat. #492016 |
| Aprotinin | Sigma | Cat. # A1153 |
| Leupeptin | Roche | Cat. #11017128001 |
| Phenylmethylsulfonyl fluoride | Sigma | Cat. #P-7626 |
| Sodium fluoride | Sigma | Cat. #S-1504 |
| Sodium orthovanadate | Sigma | Cat. #S-6508 |
| Pefabloc | Sigma | Cat. #76307-500MG |
| β-Glycerophosphate | Sigma | Cat. #G-6251 |
| **Oligonucleotides and other sequence-based reagents** | | |
| *CARD14_forward* | Integrated DNA Technologies | acccacacctggattatgagct |
| *CARD14_reverse* | Integrated DNA Technologies | tagaatgagtcccccgaggt |
| *Villin-cre_forward* | Integrated DNA Technologies | acaggcactaagggagccaatg |
| *Villin-cre_reverse* | Integrated DNA Technologies | attgcaggtcagaaagaggtcacag |
| *Villin-cre_reverse* | Integrated DNA Technologies | gttcttgcgaacctcatcactc |
| *Defa6_forward* | Integrated DNA Technologies | ctaggccacagaattgaaagatct |
| *Defa6_reverse* | Integrated DNA Technologies | gtaggtggaaattctagcatcatc |

| Reagent/resource | Reference or source | Identifier or catalog number |
|---|---|---|
| Internal control_forward | Integrated DNA Technologies | tagctgacttaaggtgcgct |
| Internal control_reverse | Integrated DNA Technologies | tcttcaaaccaagtgccctg |
| Wt control_forward | Integrated DNA Technologies | tgaaggacatctccccgccc |
| Wt control_reverse | Integrated DNA Technologies | gacagggccttctccaccc |
| MALT1_forward | Integrated DNA Technologies | gtttctcaggtctttagttcatgtc |
| MALT1_reverse | Integrated DNA Technologies | tatactctacatctccatggt |
| MALT1_reverse | Integrated DNA Technologies | ttgttttgcagatctctgcc |
| TNF^emARE_forward | Integrated DNA Technologies | tctcatgcaccaccatcaa |
| TNF^emARE_reverse | Integrated DNA Technologies | gcagaggttcagtgatgtag |
| Hprt1 forward | Integrated DNA Technologies | agtgttggatacaggccagac |
| Hprt1 reverse | Integrated DNA Technologies | cgtgattcaaatccctgaagt |
| Ubc forward | Integrated DNA Technologies | aggtcaaacaggaagacagacgta |
| Ubc reverse | Integrated DNA Technologies | tcacacccaagaacaagcaca |
| Actβ forward | Integrated DNA Technologies | gcttctaggcggactgttactga |
| Actβ reverse | Integrated DNA Technologies | gccatgccaatgttgtctcttat |
| hCARD14 forward | Integrated DNA Technologies | gtcaacacggacggttataaga |
| hCARD14 reverse | Integrated DNA Technologies | gttgacccggatgtagaatgag |
| Ccl20 forward | Integrated DNA Technologies | gtactgctggctcacctctg |
| Ccl20 reverse | Integrated DNA Technologies | cttcatcggccatctgtcttgtg |
| Tnf forward | Integrated DNA Technologies | accctggtatgagcccatatac |
| Tnf reverse | Integrated DNA Technologies | acacccattcccttcacagag |
| Nos2 forward | Integrated DNA Technologies | cagctgggctgtacaaacctt |
| Nos2 reverse | Integrated DNA Technologies | cattggaagtgaagcgtttcg |
| Il1β forward | Integrated DNA Technologies | cacctcacaagcagagcacaag |
| Il1β reverse | Integrated DNA Technologies | gcattagaaacagtccagcccatac |
| Il36γ forward | Integrated DNA Technologies | tcctgactttggggaggtttt |
| Il36γ reverse | Integrated DNA Technologies | tcacgctgactggggttact |
| Il23 forward | Integrated DNA Technologies | cccgtatccagtgtgaagatg |

| Reagent/resource | Reference or source | Identifier or catalog number |
|---|---|---|
| Il23 reverse | Integrated DNA Technologies | gggctatcagggagtagagca |
| Cxcl1 forward | Integrated DNA Technologies | gagcctctaaccagttccag |
| Cxcl1 reverse | Integrated DNA Technologies | tgagtgtggctatgacttcg |
| Cxcl2 forward | Integrated DNA Technologies | acagaagtcatagccactctc |
| Cxcl2 reverse | Integrated DNA Technologies | ttagccttgcctttgttcag |
| Defa1 forward | Integrated DNA Technologies | tcaagaggctgcaaaggaagagaac |
| Defa1 reverse | Integrated DNA Technologies | tggtctccatgttcagcgacagc |
| Lyz1 forward | Integrated DNA Technologies | gccaaggtctaacaatcgttgtgagttg |
| Lyz1 reverse | Integrated DNA Technologies | cagtcagccagcttgacaccacg |
| Gcg forward | Integrated DNA Technologies | cacgcccttcaagacacag |
| Gcg reverse | Integrated DNA Technologies | cacgcccttcaagacacag |
| Pyy forward | Integrated DNA Technologies | acggtcgcaatgctgctaat |
| Pyy reverse | Integrated DNA Technologies | aaggggaggttctcgctgtc |
| Ckk forward | Integrated DNA Technologies | tgatttccccatccaaagc |
| Ckk reverse | Integrated DNA Technologies | gcttctgcagggactaccg |
| ApoB forward | Integrated DNA Technologies | gcatgagtatgccaatggtctcc |
| ApoB reverse | Integrated DNA Technologies | ctggttgccatctgaagccatg |
| Fabp6 forward | Integrated DNA Technologies | ccccaactatcaccagacttc |
| Fabp6 reverse | Integrated DNA Technologies | acatccccgatggtggagat |
| **Chemicals, enzymes, and other reagents** | | |
| 4 kDa FITC-dextran | Sigma-Aldrich | Cat. #FD4 |
| 70 kDa FITC-dextran | Sigma-Aldrich | Cat. #46945 |
| Methylcellulose | Sigma | Cat. #M0262 |
| Carmine red dye | Sigma | Cat. #C-1022 |
| Tamoxifen | Sigma-Aldrich | Cat. #T-5648 |
| Cornoil | Sigma-Aldrich | Cat. #C8267 |
| MLT-827 | Galapagos n.v | Cat. #MLT-827 |
| Kolliphor® HS 15 | Sigma | Cat. #42966 |
| Nalidixic acid | Sigma-Aldrich | Cat. #N4382 |
| Brain Heart Infusion Agar | Sigma | Cat. #53286 |
| Ciprofloxacin | Sigma | Cat. #17850-5G-F |
| Ampicillin | Sigma | Cat. #A-9518 |
| Metronidazole | Sigma | Cat. #M-1547 |

| Reagent/resource | Reference or source | Identifier or catalog number |
|---|---|---|
| Vancomycin | Duchefa | Cat. #V0155-5 |
| Carbachol | Sigma-Aldrich | Cat. #C4382 |
| HBSS | Gibco | Cat. #14185-045 |
| EDTA | Vel | Cat. #9249 |
| DTT | Calbiochem | Cat. #233155 |
| TRIzol | Invitrogen | Cat. #15596-018 |
| RPMI | Gibco | Cat. #11875093 |
| Collagenase VIII | Sigma | Cat. #C2139 |
| ACK lysis buffer | Gibco | Cat. #A1049201 |
| PBS | Gibco | Cat. #14190-094 |
| Citrate | Vector Laboratories | Cat. #H-3300 |
| DAB substrate | Vector Laboratories | Cat. #SK-4105 |
| Entellan | Merck Millipore | Cat. #MERC1.07961.0100 |
| Polyvinyl alcohol | Sigma | Cat. #10981 |
| MyTaq DNA polymerase | Bioline | Cat. #BIO-21105 |
| BioStabII PCR Enhancer | Sigma | Cat. #53833 |
| AMPure XP beads | Agencourt | Cat. #A63881 |
| **Software** | | |
| FLUOstar Omega | BMG LABTECH | |
| Clamp Software Version 9 | Molecular Devices | |
| Chart v5.5.6 | ADInstruments | |
| Zen Lite | Zeiss Microscopy. | |
| QuPath | Bankhead et al, 2017 | |
| qBase+ | Biogazelle | |
| FlowJo | FlowJo LLC, a BD Company (Tree Star Inc) | |
| FastQC, v0.11.9 | Andrews, 2010 | |
| HISAT2 Alinger (v2.1.0) | Kim et al, 2015 | |
| R | https://www.R-project.org | |
| Bioconductor | https://www.bioconductor.org | |
| Ingenuity Pathway Analysis | QIAGEN Inc | |
| Mothur software | Schloss et al, 2009 | |
| Genstat v21 | VSN International | |
| **Other** | | |
| *C. rodentium* | KJ Maloy, University of Glasgow | ICC169 |
| EnzChek™ Lysozyme Assay Kit | Invitrogen | Cat. #E22013 |
| ImmPRESS® HRP Goat Anti-Rabbit IgG Polymer Kit | Vector Laboratories | Cat. #MP-7451 |
| PAP pen | Abcam | Cat. #ab2601 |

| Reagent/resource | Reference or source | Identifier or catalog number |
|---|---|---|
| ProLong Gold Antifade | ThermoFisher Scientific | Cat. #P36930 |
| Mm-Card14 | ACDBio | Cat. #ICC169 476041 |
| Aurum total RNA mini kit | Bio-Rad Laboratories N.V | Cat. #732-6820 |
| SensiFast cDNA synthesis Kit | GC Biotech | Cat. #BIO-650504 |
| SensiFast SYBR No-Rox kit | GC Biotech | Cat. #CSA-01190 |
| PierceTM BCA Protein Assay Kit | ThermoFisher Scientific | Cat. #23225 |
| Nitrocellulose membranes | Protran, Perkin Elmer | Cat. #NBA085A001EA |
| Western Lightning ECL detection system | Perkin Elmer | Cat. #NEL104001EA |
| 123 count eBeads | Invitrogen | Cat. #01-1234-42 |
| RNeasy Plus micro kit | QIAGEN | Cat. #74134 |
| DNeasy Powersoil Pro Kit | Qiagen | Cat. #47014 |
| Ovation Rapid DR Multiplex System 1-96 | NuGEN | Cat. #0328 |
| Illumina NovaSeq 6000 | Illumina NovaSeq | |
| Illumina MiSeq | Illumina | |

## Mice

For all experiments, 8–12-week-old littermates of both sexes were used. The mice were maintained under specific pathogen-free conditions and housed in individually ventilated cages with 12 h light/dark cycle. Mice were given ad libitum access to sterile food and autoclaved water. All experiments were performed in accordance with institutional, national, and European animal guidelines, and animal protocols were approved by the ethical committee for animal welfare at the Faculty of Sciences of Ghent University (LP0038, EC2021-010, EC2023-074, EC2023-109, EC2024-031, EC2024-034, ECD24-65K, ECD24-70). Mice were randomly assigned to different (treatment) groups. A minimum of three mice was analyzed in each group, according to the availability of the correct genotypes. During the analysis, researchers were not specifically blinded to the experimental group. All observations were retained during statistical analysis (no data points were excluded).

*CARD14*(E138A)[IEC] mice that express human *CARD14*(E138A) in IEC were generated by breeding Rosa26LSL-*CARD14*-E138A transgenic mice that express the human *CARD14*(E138A) cDNA preceded by a loxP-STOP-loxP stop cassette under the control of the ROSA26 promoter (van Nuffel et al, 2020) with Villin-Cre mice (The Jackson Laboratory). For the generation of i*CARD14*(E138A)[IEC] mice, Rosa26LSL-*CARD14*-E138A transgenic mice were bred with Villin-Cre-ERT2 mice (provided by Sylvie Robine, Institute Curie, Paris). Littermates that only express Villin-Cre or Vilin-Cre-ERT2 transgenes were used as controls. To establish *CARD14*(E138A)[PC] mice, Rosa26LSL-*CARD14*-E138A

transgenic mice were crossed with Defa6-Cre mice (provided by Richard S. Blumberg, Harvard Medical School, Boston). CARD14(E138A)^IEC mice that are MALT1-deficient mice were generated by crossing CARD14(E138A)^IEC mice with mice containing a floxed (fl) Malt1 allele that are available in our laboratory (Demeyer et al, 2019). Originally, Malt1^tm1a(EUCOMM) Hmgu/+ mice were derived from ES cells purchased from EUCOMM. Malt1^tm1a mice were further crossed with FLP deleter mice (C57BL6/J background; Rodríguez et al, 2000) to remove the FRT-flanked LacZ and neomycin selection cassette, generating the Malt1^tm1a floxed allele. TNF^emARE mice were previously described (Thiran et al, 2023). Germ-free CARD14(E138A)^IEC mice were generated via embryo transfer into axenic recipients at the germ-free and gnotobiotic mouse core facility of our center. The mice were housed in positive-pressure, flexible-film isolators (North Kent Plastics).

## Genotyping

The CARD14 transgene was genotyped using the following primers: forward ACCCACACCTGGATTATGAGCT, reverse TAGAATGAGTCCCCCGAGGT, producing a product size of 400–500 bp. Villin-cre was genotyped using forward ACAGGCACTAAGGGAGCCAATG, reverse ATTGCAGGTCAGAAAGAGGTCACAG, and reverse GTTCTTGCGAACCTCATCACTC, giving a product size of around 400 bp for Villin-Cre and 800–900 bp for the internal control. Defa6 primers were: forward CTAGGCCACAGAATTGAAAGATCT, reverse GTAGGTGGAAATTCTAGCATCATC (internal control); forward TAGCTGACTTAAGGTGCGCT, reverse TCTTCAAACCAAGTGCCCTG (WT control); forward TGAAGGACATCTCCCGCCC, reverse GACAGGGCCTTCTCCACCC (Cre), giving product sizes of 425, 250, and 200 bp, respectively. MALT1 floxed allele was genotyped using forward GTTTCTCAGGTCTTTAGTTCATGTC, reverse TATACTCTACATCTCCATGGT, and reverse TTGTTTTGCAGATCTCTGCC, resulting in 280, 448, and 345 bp for WT, floxed, and KO, respectively, TNF^emARE forward TCTCATGCACCACCATCAA, TNF^emARE reverse GCAGAGGTTCAGTGATGTAG, giving product of 390 bp and 500 bp for WT.

## Intestinal permeability

Eight-week-old mice were fasted for 4 h before orally administering 4 kDa FITC-dextran (0.5 mg/g body weight in PBS, FD4, Sigma-Aldrich). Blood was collected 2 h later via tail bleeding, incubated for 30 min at RT, and serum was separated by centrifugation at 3000 rpm for 30 min at 4 °C. Fluorescence was measured in 96-well plates at excitation wavelength 485 nm and emission wavelength 520 nm using FLUOstar Omega (BMG LABTECH). Concentrations were calculated using a standard range between 125 and 8000 ng/ml. Mouse serum without 4 kDa FITC-dextran administration was used to subtract the background.

## Gastrointestinal transit time and gastric emptying

A gavage solution was prepared by dissolving methylcellulose (0.5% w/v, M0262, Sigma) in hot distilled water and mixed with carmine red dye (6%, C-1022, Sigma). Each mouse was placed in a separate cage without bedding and 200 μL of the gavage solution was orally administered. Mice were monitored every 15–20 min for the appearance of the first colored pellet. The interval between gavage and the first appearance of red pellet is the whole gut transit time. For the measurement of gastric emptying, mice were gavaged with non-absorbable 70 kDa FITC-dextran (5 mg/ml, 46945, Sigma-Aldrich) dissolved in methylcellulose (2%, M0262, Sigma), and the percentage of total fluorescence signal remaining in the stomach 5 min after oral gavage was measured using FLUOstar Omega. Non-fasted mice were used for the experiments.

## Neutrophil depletion

In all, 8–9-week-old iCARD14(E138A)^IEC mice were administered 5 mg Tamoxifen (T-5648, Sigma-Aldrich) dissolved in corn oil (C8267, Sigma-Aldrich) via oral gavage on day 1. The mice were treated intraperitoneally with either 200 μg control antibody (BE0090, BioxCell) or 200 μg anti-Gr-1 antibody (BE0075, BioxCell) on days 1 and 3. Gastrointestinal transit time was measured on day 0 before tamoxifen administration and again on day 5. Neutrophils in the blood and lamina propria of the small intestine were analyzed by flow cytometry.

## MALT1 inhibition

In all, 8–9-week-old iCARD14(E138A)^IEC mice were administered 5 mg Tamoxifen (T-5648, Sigma-Aldrich) dissolved in corn oil (C8267, Sigma-Aldrich) via oral gavage on days 1, 2, and 3. For MALT1 inhibitor treatment, mice received a daily intraperitoneal injection (i.p.) of 30 mg/kg MALT1 inhibitor (MLT-827; provided by Galapagos n.v., Mechelen) for 7 days. The MALT1 inhibitor was prepared by dissolving it in a Kolliphor® HS 15 (42966, Sigma)/0.5% methylcellulose (M0262, Sigma) solution at 2:98 ratio under continuous stirring and protected from light. Control mice were treated with vehicle solution (Kolliphor® HS 15/0.5% methylcellulose, 2:98 ratio).

## TNF neutralization

In all, 8–9-week-old iCARD14(E138A)^IEC mice received a single oral dose of 5 mg Tamoxifen (T-5648, Sigma-Aldrich) dissolved in corn oil (C8267, Sigma-Aldrich) on day 1. Mice were treated intraperitoneally with either anti-TNF antibody (200 μg, BE0058, BioXCell) or PBS on days 1 and 4.

## Citrobacter rodentium infection

Age- and sex-matched mice were infected with 5 × 10^9 CFU of the nalidixic acid (N4382, Sigma-Aldrich) resistant ICC169 C. rodentium strain administered by oral gavage in a 200 μl inoculum during the logarithmic phase of proliferation. Body weight was monitored every other day and C. rodentium loads were determined by plating stool samples on selective LB agar containing 50 μg/ml nalidixic acid every 3 days post infection. Colony-forming units (CFUs) were normalized to sample weight.

## Depletion of gut microbiota by antibiotics

To deplete intestinal microbiota, 8–12-week-old iCARD14(E138A)^IEC mice were given broadspectrum antibiotics (200 mg/l ciprofloxacin, 17850-5G-F, Sigma; 1 g/l ampicillin, A-9518, Sigma; 1 g/l metronidazole,

M-1547, Sigma; 500 mg/l vancomycin, V0155-5, Duchefa) in their drinking water for 3 weeks. The drinking water was changed twice weekly. On days 7, 14, and 21, fecal samples were collected in sterile 2-ml tubes containing sterile PBS and beads, and lysed using Tissue Lyser II (Qiagen). In total, 100 µl of the fecal lysate was plated on brain heart infusion agar plates (53286, Sigma) and incubated at 37 °C overnight. Untreated mice were used as a positive control. Microbiome depletion was evaluated by the absence of colonies on the plates. On day 22, tamoxifen was administered to induce CARD14 expression, and on day 27, GIT and stool collection were performed.

## Ussing chambers

Mice were sacrificed by cervical dislocation. Immediately after, the ileum was carefully removed to prevent tissue damage and maintained in oxygenated Krebs solution (95% $O_2$–5% $CO_2$). Intestinal samples were then pinned in Sylgard plates and the mucosal layer was gently peeled off to prepare a lamina propria-submucosal plexus intestinal strip. These intestinal preparations were mounted in Ussing chambers (exposed area 0.0096 cm$^2$) and the mucosal and serosal compartments were bathed independently with 3 ml Krebs buffer (pH 7.4) containing 118 mM NaCl, 4.7 mM KCl, 1.2 mM $CaCl_2$, 1.2 mM $MgSO_4$, 1.2 mM $NaH_2PO_4$, 25 mM $NaHCO_3$). Glucose (10 mM) was present in the medium that bathed the serosal surface of the tissue and mannitol (10 mM) was substituted for glucose in the medium that bathed the mucosal surface. Media were continuously oxygenated (95% $O_2$ − 5% $CO_2$) and maintained at 37 °C throughout the experiment. After mounting, the tissue was allowed to equilibrate for 30 min. Thereafter, tissues were voltage-clamped at 0 mV using an automatic voltage clamp, and the short-circuit current ($I_{sc}$) to keep the 0-mV potential was monitored over time. TEER was calculated by using Ohm's law. After short-circuiting, intestinal tissue was equilibrated for 10-15 min to allow bioelectric measures to stabilize, and veratridine (30 µM) was added to the basolateral reservoir to stimulate voltage-sensitive $Na^+$ intrinsic submucosal neurons (Sheldon et al, 1990). Veratridine-induced changes in $I_{sc}$ were recorded for 60 min. Chambers were connected to a computer with Clamp Software Version 9 (KMSCI) to record $I_{sc}$.

## Muscle strips–organ bath experiments

Ileal segments were freshly isolated and maintained in oxygenated Krebs solution (95% $O_2$ − 5% $CO_2$). The samples were opened along the mesenteric border, pinned flat in a Sylgard-lined dish, and the mucosal layer was carefully removed. Strips were cut (5 mm × 20 mm) and suspended longitudinally in organ bath chambers filled with constantly oxygenated Krebs solution at 37 °C. Strips were mounted at an initial tension of 0.5 g of tissue and washed every 15 min over the course of 1 h. After the equilibration was reached, electric field stimulation was applied via a 4-channel custom-made stimulator (TOWO, Ghent University, Belgium) with the following parameters: 2–16 Hz, 0.5 ms, 45 V in trains of 10 s. The contractile responses to electric field stimulation were recorded, and the maximal contractile response was noted. After each stimulus, the muscle strips were washed at least three times every 5 min. Afterward, tissues were exposed to cumulative concentrations of carbachol (C4382, Sigma-Aldrich) to directly activate muscarinic receptors on muscle tissue. Every 1.5 min after

tissue reached the maximal contraction, the carbachol concentration was increased ten times ($10^{-9}$–$10^{-2}$ M). Changes in isometric force were recorded by MLT050/D force transducers connected to an 8-channel Powerlab/8SP ML785 data acquisition system coupled to a computer containing Chart v5.5.6 software (ADInstruments, United Kingdom). All contractile responses were normalized to the weight of the tissue.

## Transmission electron microscopy

Mouse ileum and colon samples were cut into small pieces and immersed in a fixative solution of 2.5% glutaraldehyde and 3% formaldehyde in 0.1 M Sodium cacodylate buffer, placed in a vacuum oven for 30 min, and left rotating for 3 h at room temperature. This solution was later replaced with fresh fixative, and samples were left rotating overnight at 4 °C. After washing, samples were post fixed in 1% $OsO_4$ with $K_3Fe(CN)_6$ in 0.1 M Sodium cacodylate buffer, pH 7.2. After washing, samples were subsequently dehydrated through a graded ethanol series, including a bulk staining with 2% uranyl acetate at the 50% ethanol step, followed by embedding in Spurr's resin. To select the area of interest on the block and in order to have an overview of the phenotype, semi-thin sections were first cut at 0.5 µm and stained with toluidine blue. Ultrathin sections of a gold interference color were cut using an ultramicrotome (Leica EM UC6), followed by a post-staining in a Leica EM AC20 for 40 min in uranyl acetate at 20 °C and for 10 min in lead stain at 20 °C. Sections were collected on formvar-coated copper slot grids. Grids were viewed with a JEM-1400Plus transmission electron microscope (JEOL, Tokyo, Japan) operating at 80 kV.

## Isolation of IEC

The small intestine and colon were isolated and thoroughly flushed with cold PBS and placed in 1× HBSS (14185-045, Gibco) solution supplemented with 2% FCS. Peyer's patches were removed from the small intestine, and fat was removed from both the small intestine and the colon. The tissue was turned inside out and placed into the tube with 1× HBSS solution supplemented with 30 mM EDTA (9249, Vel) and 2 mM DTT (Cat. #233155, Calbiochem) on ice for 30 min. After incubation, the tubes were vigorously shaken to release IEC. The samples were filtered through a 70-µm strainer and spun down at 400 × $g$ for 5 min and resuspended in TRIzol (15596-018, Invitrogen) for subsequent RNA isolation.

## Isolation of intestinal lamina propria

The small intestine was turned inside out and placed into a tube with 1× HBSS solution supplemented with 5 mM EDTA for 20 min at 37 °C under continuous shaking at 120 rpm. Afterward, the tissue was washed in warm 1× HBSS and placed into a tube with fresh 5 mM EDTA-HBSS solution for another 20 min incubation. The tissue was then washed to remove any remaining EDTA and placed into a 2-ml tube with RPMI (11875093, Gibco). The tissue was finely cut with scissors inside a 2-ml tube and transferred into a 50 ml tube containing a digestive cocktail (0.6 mg/ml Collagenase VIII (C2139, Sigma) in RPMI (11875093, Gibco)). The tubes were placed into a water bath at 37 °C for 10-15 min under continuous shaking at 120 rpm. To stop the enzymatic reaction, 1× PBS

supplemented with 10% FCS and 2 mM EDTA was added to the samples. Samples were filtered through a 70-μm strainer and spun down at $400 \times g$ for 5 min. Pellets were resuspended in 1× PBS and transferred into a FACS plate.

## Whole blood isolation

Blood was collected from each mouse via cardiac puncture into tubes containing 0.5 M EDTA. The blood samples were lysed in ACK lysis buffer (A1049201, Gibco), vortexed, incubated for 3 min at room temperature, centrifuged, and transferred into a FACS plate.

## Lysozyme activity measurement

Ileum samples were collected, snap frozen, and stored at $-70\,°C$. For analysis, tissues were homogenized using Precellys 24 (Bertin Technologies with CK26 beads) in 1× PBS (14190-094, Gibco). Lysozyme activity was measured using EnzChek™ Lysozyme Assay Kit, which contains fluorescent-labeled peptidoglycan from *Micrococcus lysodeikticus* (E22013, Invitrogen). Lysed samples were incubated with the lysozyme substrate at $37\,°C$ for 30 min. Fluorescence signal (excitation/emission of 485/520 nm) was measured by FluorSTAR, and lysozyme activity was determined from a standard curve. Assays were conducted with lysozyme concentrations ranging from 0.0001 to 2.5 mg/ml.

## TNF protein level

Ileal tissue was lysed in RIPA buffer (1 M Tris pH 7.6, 5 M NaCl, 0.5% NP-40 Alternative, 0.5% deoxycholic acid, 10% SDS, 0.5 M EDTA) with following protease and phosphatase inhibitors: 0.15 μM aprotinin (A1153, Sigma), 2.1 μM leupeptin (11017128001, Roche), 0.1 mM phenylmethylsulfonyl fluoride (P-7626, Sigma), 10 mM sodium fluoride (S-1504, Sigma), 1 mM sodium orthovanadate (S-6508, Sigma), 1 mM pefabloc (76307, Sigma), 0.175 mM β-glycerophosphate (G-6251, Sigma). TNF protein levels were quantified using a TNF ELISA kit (88-7324-88, Invitrogen) following the manufacturer's instructions. Total protein concentrations were determined using Pierce™ BCA Protein Assay Kit (23225, Thermo-Fisher Scientific) and used for normalization.

## Tissue processing and histological staining

Small intestine and colon tissue were fixed in 4% paraformaldehyde in PBS overnight at $4\,°C$, followed by dehydration and embedding in paraffin. Sections of 5 μm were deparaffinized and then stained with hematoxylin and eosin or Alcian Blue–Periodic Acid–Schiff (AB-PAS) using an Autostainer XL (Leica). After dehydration, tissues were mounted with Entellan mounting medium (MERC1.07961.0100, Merck Millipore) and stored at room temperature until image acquisition. Whole-slide images were acquired using the Axio Scan.Z1 slide scanner (Zeiss) and analyzed with Zen Lite software (Zeiss).

## Immunohistochemistry and immunofluorescence

### Chromogenic IHC

For chromogenic immunohistochemistry, tissue sections were deparaffinized, rehydrated, and antigen retrieval was performed by heating in citrate-based antigen unmasking solution for 20 min

(H-3300, Vector Laboratories). Endogenous peroxidase activity was quenched with 3% $H_2O_2$ in methanol. Non-specific binding was blocked using 5% normal goat serum with 1% BSA in PBS. Sections were incubated overnight at $4\,°C$ with anti-CD45 primary antibody (1:10,000; ab10558, Abcam). The next day, sections were incubated with the ImmPRESS® HRP Goat Anti-Rabbit IgG Polymer Kit (1:5, MP-7451, Vector Laboratories), developed using DAB substrate (ImmPACT DAB, SK-4105, Vector Laboratories), and counterstained with hematoxylin. Slides were mounted with Entellan (MERC1.07961.0100, Merck Millipore) imaged with the Axio Scan.Z1 (Zeiss), and analyzed with Zen Lite (Zeiss) and QuPath software.

### Immunofluorescence (lysozyme)

For fluorescent-based staining, sections were also deparaffinized, rehydrated, and heated in citrate-based antigen unmasking solution (H-3300, Vector Laboratories) in a 2100-retriever pressure cooker (PickCell Laboratories). To reduce autofluorescent background, sections were incubated with 0.5% $NaBH_4$ in PBS for 30 min at room temperature. Slides were then washed three times 5 min in PBS, and regions of interest were circled with a PAP pen (ab2601, Abcam). Non-specific binding was blocked with 5% goat serum in PBT (PBS + 0.5% BSA + 0.1% Tween-20) for 30 min at room temperature. Followed by overnight incubation with the primary anti-lysozyme antibody (1:1000, A0099, Dako) at $4\,°C$. The next day, sections were incubated for 2 h at room temperature with goat anti-rabbit AF 568 secondary antibody (1:500, ab175471, Abcam) in the dark. Nuclei were counterstained with DAPI (1:1000) for 15 min at room temperature, rinsed and mounted with polyvinyl alcohol (10981, Sigma), dried overnight, and imaged using Axio Scan.Z1 slide Scanner (Zeiss). Images were analyzed by using Zen Lite (Zeiss) and quPath software.

## Neuron staining

Mice were euthanized by CO2 overdose, and the small intestine was removed and placed in ice-cold HBSS (GIBCO) with 5% FBS. The small intestine was flushed with ice-cold PBS to remove luminal contents. For whole-mount immunofluorescence, the intestine was cut open longitudinally and stretched on a Sylgard plate. The muscularis externa was carefully removed from the remaining submucosa and lamina propria. Intestinal whole-mount tissues were fixed for 30 min in 4% PFA and then extensively washed with PBS before permeabilizing in 0.3% Triton X-100 for 2 h at room temperature and blocking for 2 h in 5% BSA and 5% Donkey serum in 0.5% Triton X-100 at room temperature. Subsequently, samples were incubated overnight at $4\,°C$ with the primary Anti-HuC/D antibody (provided by V. Lennon, Mayo Clinic) in blocking buffer. Samples were washed in PBS and incubated with secondary Cy3-conjugated donkey anti-human (Jackson ImmunoLabs) antibody in blocking buffer. Subsequently, samples were washed in PBS and mounted on slides with ProLong Gold Antifade (ThermoFisher Scientific). Microscopy was performed with Axio Scan.Z1 slide Scanner (Zeiss), ×10 objective. Images were analyzed using ImageJ and QuPath software. A ganglion was defined as a cohesive aggregate of HuC/D+ cells. At least 100 ganglia per mouse were counted. Extra-ganglionic cells were not counted.

## RNAscope

Tissue samples were fixed in 10% neutral buffered formalin for 48 h at room temperature, and kept stored in 70% ethanol. The tissues were embedded in paraffin, and sections (5 μm) were cut. For immunohistochemistry, tissue sections were dewaxed three times in xylene for 1 min each. The slides were then rehydrated with a series of decreasing concentrations of ethanol (1-min incubations) ending with distilled water. For *Card14* mRNA detection in tissue, RNAscope Probe - Mm-Card14 (Cat No 476041, ACDBio) was used according to the manufacturer's instructions.

## RNA extraction, cDNA synthesis, and quantitative real-time PCR

IEC were lysed in TRIzol reagent (15596-018, Invitrogen), followed by phenol-chloroform phase separation. RNA extraction was then performed using Aurum total RNA mini kit (732-6820, Bio-Rad Laboratories N.V.). cDNA was synthesized using a SensiFast cDNA synthesis Kit (BIO-650504, GC Biotech) according to the manufacturer's instructions. Quantitative PCR was performed using LightCycler 480 (Roche) with SensiFast SYBR No-Rox kit (CSA-01190, GC Biotech) using a total of 10 ng cDNA and 1 μM of specific primer in a total volume of 10 μl. Samples were analyzed in triplicate. Analysis was done using qBase+ software (Biogazelle, Gent, Belgium). Values were normalized to the appropriate amount of reference genes, as determined by geNorm analysis in the qBase software. The following primers were used: *Hprt1* forward, AGTGTTGGATACAGGCCAGAC; *Hprt1* reverse, cgtgatt-caaatccctgaagt; *Ubc* forward, aggtcaaacaggaagacagacgta; *Ubc* reverse, tcacacccaagaacaagcaca; *Actb* forward, GCTTCTAGGCG-GACTGTTACTGA; *Actb* reverse, gccatgccaatgttgtctcttat; *hCARD14* forward, gtcaacacggacggttataaga; *hCARD14* reverse, gttgacccggatgtagaatgag; *Ccl20* forward, gtactgctggctcacctctg; *Ccl20* reverse, cttcatcggccatctgtcttgtg; *Tnf* forward, accctggtgatgagcccatatac; *Tnf* reverse, acacccattcccttcacagag; *Nos2* forward cagctgggctgta-caaacctt, *Nos2* reverse cattggaagtgaagcgtttcg, *Il1b* forward cacctca-caagcagagcacaag, *Il1b* reverse gcattagaaacagtccagcccatac, *Il36g* forward tcctgactttgggggaggtttt, *Il36g* reverse tcacgctgactggggttact, *Il23* forward cccgtatccagtgtgaagatg, *Il23* reverse gggctatcagggagta-gagca, *Cxcl1* forward gagcctctaaccagttccag, *Cxcl1* reverse tgagtgtggctatgacttcg, *Cxcl2* forward acagaagtcatagccactctc, *Cxcl2* reverse ttagccttgcctttgttcag *Defa1* forward, tcaagaggctgcaaaggaaga-gaac, *Defa1* reverse, tggtctccatgttcagcgacagc, *Lyz1* forward, gccaaggtctaacaatcgttgtgagttg, *Lyz1* reverse, cagtcagccagcttgacaccacg, *Gcg* forward cacgcccttcaagacacag, *Gcg* reverse cacgcccttcaagacacag, *Pyy* forward acggtcgcaatgctgctaat, *Pyy* reverse aaggggaggttctcgctgtc, *Ckk* forward tgatttccccatccaaagc, *Ckk* reverse gcttctgcagggactaccg, *Apob* forward gcatgagtatgccaatggtctcc, *Apob* reverse ctggttgccatct-gaagccatg, *Fabp6* forward ccccaactatcaccagacttc, *Fabp6* reverse acatcccgatggtggagat.

## Western blotting

IEC were lysed in colorless Laemmli buffer (50 mM Tris–HCl pH 8, 2% SDS, 10% glycerol). Protein concentrations were determined using PierceTM BCA Protein Assay Kit (23225, ThermoFisher Scientific), and equal amounts of protein extract were separated by 10% SDS–PAGE. Proteins were then transferred to nitrocellulose membranes (0.45-μm pore size; Protran, Perkin Elmer). Membranes were blocked in 5% milk powder in TBS / 0.2% Tween-20 (TBST) for 1 h at room temperature and probed with specific primary antibodies in 5% milk powder in TBST overnight at 4 °C. The following antibodies were used: anti-CARD14 (1:1000, 10400-1-AP; Proteintech), anti-CARD14 (1:1000, NBP2-92873; Novus Biologicals), anti-CYLD (1:1000, sc-74435, Santa Cruz), anti-actin monoclonal antibody (1:10,000, MP6472J; MP Biomedicals), secondary HRP conjugated anti-rabbit IgG antibody and anti-mouse IgG antibody (1:5000, 31432 and 31464, respectively; ThermoFisher Scientific). Bound secondary antibodies were detected using the Western Lightning ECL detection system (NEL104001EA, Perkin Elmer) according to the manufacturer's instructions.

## Flow cytometry

Flow cytometry analysis was performed on LSR Fortessa 4 lasers (BD Biosciences) at the VIB Flow Core facility. Total lamina propria and blood cell suspensions were pre-incubated with purified rat anti-mouse CD16/CD32 (1:400, 553142, BD Bioscience). Cells were incubated with fixable viability dye eFluor506 (1:200, 65-0866-18, eBioscience) and the following antibodies: anti-CD45-FITC (1:800, 11-0451-82, Invitrogen), anti-CD11b-V450 (1:200, 560455, BD Bioscience), CD64-BV711 (1:100, 139311, Biolegend), CD3-PeCy5 (1:200, 550031,Tonbo Biosciences), CD19-PeCy5 (1:400, 15-0193, eBioscience), SiglecF-PE (1:1000, 552126, BD Bioscience), CD11c-PeCy7 (1:400, 117317, Biolegend), Ly6C-AlexaFluor700 (1:300, 561236, BD), MHCII-APCeFluor780 (1:800, 47-5321-82, eBioscience), and Ly6G-APC (1:800, 127613, Biolegend). 123 count eBeads (01-1234-42, Invitrogen) were used for cell quantification. Final analysis was performed using FlowJo software (Tree Star Inc.). Different cell populations were gated as follows: neutrophils (CD45$^+$CD3/CD19$^-$CD11b$^+$Ly6G$^+$), eosinophils (CD45$^+$CD3CD19$^-$Ly6G$^-$CD11b$^+$SiglecF$^+$), dendritic cells (CD45$^+$CD3/CD19$^-$Ly6G$^-$CD11b$^+$Ly6C$^-$CD64$^-$CD11c$^+$MHCII$^+$), T cells (CD45$^+$ CD3CD19$^+$MHCII$^-$) and B cells (CD45$^+$CD3CD19$^+$MHCII$^+$).

## Bulk RNA sequencing of IEC

IEC were isolated as described above. Total RNA was extracted using the RNeasy Plus Micro Kit (QIAGEN, 74134) according to the manufacturer's instructions. RNA concentration and purity were determined spectrophotometrically using the Nanodrop ND-1000 (Nanodrop Technologies), and RNA integrity was assessed with the Bioanalyzer 2100 (Agilent). All samples had RNA integrity numbers (RIN) of 7.1 or higher. For each sample, 250 ng of total RNA was used as input for library preparation. Poly-A containing mRNA molecules were purified using poly-T oligo-attached magnetic beads, followed by first-strand cDNA synthesis with random primers and second-strand synthesis using DNA Polymerase I and RNase H (Illumina TruSeq® Stranded mRNA Sample Prep Kit, protocol version #1000000040498 v00, October 2017). Blunt-ended cDNA fragments were extended with a single 'A' base and ligated to indexing adapters. Enrichment PCR was then performed to amplify fragments with adapter sequences at both ends. Libraries were equimolarly pooled and sequenced on an Illumina NovaSeq 6000 (v1.5 kit, S1 flowcell, single-end 100 bp

reads) at the VIB Nucleomics Core (https://nucleomicscore.sites.vib.be/).

Raw sequencing quality was assessed using FastQC, v0.11.9. Overall read quality was excellent, and no contamination with Illumina adapter sequences was found, so no pre-processing steps were needed. However, the base composition at the 5' end of the reads differs from the rest. For this reason, the first 15 bases were removed from all reads during the mapping step. Reads were mapped to the mouse reference genome (mm10) using the HISAT2 Alinger (v2.1.0) and counted via DESeq2. In addition, to check for CARD14 human transgene expression, unmapped reads were mapped to the human genome/transcriptome (hg19, Ensembl release 100). Data analysis was conducted using R/Bioconductor with the limma packages for normalization and differential expression analysis. DE genes were identified with adjusted $P$ value of 0.05 and the $\log_2$FC (fold change) threshold of $< -1$ or $>1$. Upregulated and downregulated genes were both analyzed with Enrichr and Ingenuity Pathway Analysis (IPA®, QIAGEN Inc.). Enrichr identified gene ontologies and pathways, whereas IPA detected top canonical pathway.

## DNA extraction from stool samples

Freshly evacuated fecal pellets were collected from 8 to 12-week-old mice, transferred to sterile 1.5-ml Eppendorf tubes, and stored at $-80\,°C$ for 1–2 months. For DNA extraction, stool pellets were weighed and dissolved in filtered PBS (PBS filtered with a 0.20-μm CHROMAFIL® Xtra PVDF-20/25 syringe filter) to a final concentration of 300 mg/ml.

DNA was extracted using the DNeasy Powersoil Pro Kit following the manufacturer's instructions with slight modifications to the mechanical cell disruption step. Briefly, 60–100 mg of homogenized fecal material was added to a bead-containing PowerBead Pro Tube and mixed with 800 μl of Solution CD1. Samples were mechanically disrupted using a PowerLyzer 24 Homogenizer (MoBio) at 2000 rpm for 5 min and then centrifuged at 15,000 relative centrifugal force for 1 min. In all, 600 μl supernatant was transferred to a clean 2-ml tube and the samples were further processed following the manufacturer's protocol. Extracted DNA was eluted and stored at $-20\,°C$ until 16S rRNA Illumina sequencing.

## Illumina MiSeq 16S rRNA gene amplicon sequencing and bioinformatics analysis

DNA extracts were sent to LGC Genomics (Teddington, Middlesex, UK) for next-generation 16S rRNA gene amplicon sequencing of the V3-V4 region (341F-785R) on an Illumina MiSeq platform using Illumina V3 chemistry (Illumina, Hayward, CA, USA), as detailed in De Paepe et al, 2017. The 16S rRNA gene V3-V4 hypervariable regions of the extracted DNA were amplified by PCR using the barcoded versions of the primers derived from Klindworth et al, 2013, with slight modification to the reverse primer by introducing another wobble position (K) to make it more universal. The PCR mix included 1 μl of DNA extract, 15 pmol of both the forward (341 F 5'-NNNNNNNNNTCCTACGGGNGGCWGCAG) and reverse primer (785 R 5'- NNNNNNNNNNTGACTACHVGGGTATCTAAKCC) in 20 μl volume of MyTaq buffer containing 1.5 units MyTaq DNA polymerase (Bioline) and 2 μl of

BioStabII PCR Enhancer (Sigma). For each sample, the forward and reverse primers had the same unique 10-nt barcode sequence. PCR conditions: initial denaturation for 2 min at 96 °C, followed by 20 cycles of 15 s denaturation at 96 °C, 30 s annealing at 50 °C and 90 s extension at 70 °C. DNA concentration of amplicons was determined by gel electrophoresis. Approximately 20 ng of amplicon DNA from each sample was pooled, purified with AMPure XP beads (Agencourt) and further purified on MinElute columns (Qiagen). Libraries were constructed using the Ovation Rapid DR Multiplex System 1-96 (NuGEN), pooled, size selected by preparative gel electrophoresis, and sequenced on an Illumina MiSeq using v3 Chemistry (Illumina).

Illumina amplicon sequencing data were processed in R version 4.0.3 using the mothur software package (v.1.40.5). In short, forward and reverse reads were combined into contigs of approximately 427 bps, aligned to the mothur-formatted silva_seed release 138 alignment database, and trimmed to match 341F-785R primers. Non-aligning sequences and sequences with homopolymer stretches >12 bases were removed, and sequences were pre-clustered allowing up to four differences. UCHIME was applied to remove chimera. Sequences were classified against the silva taxonomy database (v138_1) with an 85% confidence score and only bacterial sequences were retained and binned into Operational Taxonomic Units (OTUs) with 97% sequence similarity. The shared file containing OTU read counts and assigned taxonomy was loaded into R version 4.0.3 and converted into a phyloseq object using the phyloseq R package (v1.34.0). Rarefaction curves were constructed (package vegan_2.5–7) to ensure sufficient sequencing depth. Alpha diversity metrics were calculated using the phyloseq R package. Singletons were filtered out, and proportional read counts were converted to absolute counts using total cell counts per gram of stool. Principal Coordinate Analysis (PCoA) based on Bray-Curtis dissimilarity was conducted and visualized using the phyloseq.

## Statistical analysis

Results are expressed as the mean ± SEM. Statistical analysis between two groups was done with GraphPad Prism 9 using the unpaired $T$ test or Mann–Whitney $U$ test. $P$ values were corrected for multiple testing (where necessary) using a False Discovery Rate (FDR) approach, with the FDR value set at 5%. A conjugate Hierarchical Generalized Linear Model (HGLM), as implemented in Genstat v24 (VSN International, 2022), with fixed terms Gene*Genotype, and sample as random term, was fitted to the T cell, B cell, DC, eosinophil and neutrophil count data simultaneously (Fig. 1D). The model assumes a Poisson distribution for the fixed effects and a Gamma distribution for the random effects. Both distributions come with a log link function. The dispersion parameter for the variance of the response was estimated from the residual mean square of the fitted model. $T$ statistics were used to assess the significance of genotype effects (on the log-transformed scale) by pairwise comparisons to WT set as the reference level.

Statistical analysis between the four groups was done with GraphPad Prism 9 using two-way ANOVA. A two-way ANOVA examined the treatment and genotype factors simultaneously. If no interaction occurred, pairwise comparisons of genotype-treatment combinations were assessed by the main effects. If there was

## The paper explained

### Problem

CARD14 is an intracellular scaffold protein that is predominantly expressed in skin keratinocytes, where it has been shown to control skin inflammation. Importantly, rare human CARD14 variants have been associated with psoriasis and atopic dermatitis in patients. In addition to skin keratinocytes, CARD14 is also expressed in intestinal epithelial cells, but its function in the intestine remains unknown.

### Results

To better understand the effect of CARD14 signaling in the intestine, we generated transgenic mice that express the psoriasis-associated gain-of-function human CARD14(E138A) mutant protein specifically in intestinal epithelial cells. These mice show mild intestinal inflammation, underscoring a possible link between dysregulated CARD14 signaling and colitis. Unexpectedly, CARD14 signaling in transgenic CARD14(E138A)[IEC] mice drastically reduced intestinal transit, resulting in a constipation-like phenotype. Our studies further revealed functional changes in intestinal secretory cells, including decreased expression of antimicrobial peptides by Paneth cells, accompanied by microbial dysbiosis, which we propose to lead to impaired epithelial–neuronal communication.

### Impact

Our data support a correlation between psoriasis and subclinical intestinal disease, indicating that it might be important to increase awareness of intestinal disease symptoms among certain psoriasis patients and treating dermatologists. The mice that we have generated also provide the field with an interesting new model to study the specific role of epithelial cells in driving the pathology of intestinal motility disorders, such as constipation and irritable bowel syndrome, and to test novel therapeutics.

interaction, specific contrasts were tested for significance. If significant interaction was found, post hoc comparisons were performed with Tukey's multiple comparison test. If no interaction was detected, comparisons were made within each factor using Sidak's multiple comparison test. A log-linear regression model, as implemented in Genstat v24 (VSN International, 2022), was fitted to the gene expression data from four groups. The dispersion parameter for the variance of the response was estimated from the residual mean square of the fitted model. T statistics were used to assess the significance of genotype effects and/or treatment effects (on the log-transformed scale) by pairwise comparisons to WT and/or Control set as reference level.

Frequency and concentration-response curves in muscle strips, as well as body weight and CFU counts in feces, were analyzed as repeated measurements using the residual maximum likelihood (REML) approach, as implemented in Genstat v24 (VSN International, 2022). Some data were log-transformed, as indicated.

## Data availability

The RNA-seq dataset generated in this study has been deposited in the NCBI Gene Expression Omnibus (GEO) under accession number GSE302705. The metagenomic sequencing data have been deposited in the Sequence Read Archive (SRA) under accession number PRJNA1292125.

The source data of this paper are collected in the following database record: biostudies:S-SCDT-10_1038-S44321-025-00321-4.

## Peer review information

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

## Acknowledgements

Research in the Beyaert lab was supported by the VIB, the Research Foundation – Flanders (FWO) (G035517N, 3G086521 and G0A7T24N), an FWO-Excellence of Science Programme grant (3G0I1422), and Basic Research funding from Ghent University (bof/baf/4 y/2024/01/511). Research in the Libert lab was supported by Strategic Basic Research grants from FWO (3S003122 and 3179K5620), Ghent University grant GOA (01G00419) and Methusalem (01M00121). Research in the Bosmans lab was supported by the FWO (G000220N). RC Collaço is funded by a FWO junior postdoctoral fellowship under application 12Z3922N. IS Afonina holds a postdoctoral grant from the Stichting tegen Kanker (365C06721). We thank the VIB Flow Core, the IRC Cell Culture Facility, the IRC Transgenic Core Facility, the IRC Animal House Facility, the Germ-Free and Gnotobiotic Mouse Facility at Ghent University, and the VIB Nucleomics Core for training and technical support. We also like to acknowledge Amanda Gonçalves, Eef Parthoens, and Riet De Rycke (VIB Bioimaging Core-Ghent) for technical support and data analysis, and Marnik Vuylsteke for help with statistical analysis. We also thank other members of the Beyaert lab (Jill Steels, Femke De Meyer, Natalia Ferreras Moreno) for offering extra hands with some mouse experiments. SCL is supported by a Wellcome Trust Investigator Award (222487/Z/21/Z).

## Author contributions

**Aigerim Aidarova**: Conceptualization; Formal analysis; Validation; Investigation; Visualization; Methodology; Writing—original draft; Writing—review and editing. **Marieke Carels**: Formal analysis; Investigation; Methodology; Writing—review and editing. **Mira Haegman**: Investigation; Methodology; Writing—review and editing. **Yasmine Driege**: Investigation; Methodology; Writing—review and editing. **Steven Timmermans**: Data curation; Software; Formal analysis; Writing—review and editing. **Eline Van Damme**: Data curation; Software; Formal analysis; Investigation; Methodology; Writing—review and editing. **Javier Aguilera-Lizarraga**: Formal analysis; Investigation; Methodology; Writing—review and editing. **Maria Francesca Viola**: Investigation; Methodology; Writing—review and editing. **Rita de Cássia Collaço**: Investigation; Methodology; Writing—review and editing. **Joan Manils**: Investigation; Methodology; Writing—review and editing. **Steven C Ley**: Supervision; Writing—review and editing. **Frank Bosmans**: Supervision; Writing—review and editing. **Tom Van de Wiele**: Supervision; Writing—review and editing. **Guy Boeckxstaens**: Supervision; Writing—review and editing. **Claude Libert**: Supervision; Writing—review and editing. **Rudi Beyaert**: Conceptualization; Supervision; Funding acquisition; Writing—original draft; Project administration; Writing—review and editing. **Inna S Afonina**: Conceptualization; Formal analysis; Supervision; Funding acquisition; Investigation; Methodology; Writing—original draft; Project administration; Writing—review and editing.

Source data underlying figure panels in this paper may have individual authorship assigned. Where available, figure panel/source data authorship is listed in the following database record: biostudies:S-SCDT-10_1038-S44321-025-00321-4.

## Disclosure and competing interests statement

The authors declare no competing interests.

# Expanded View Figures

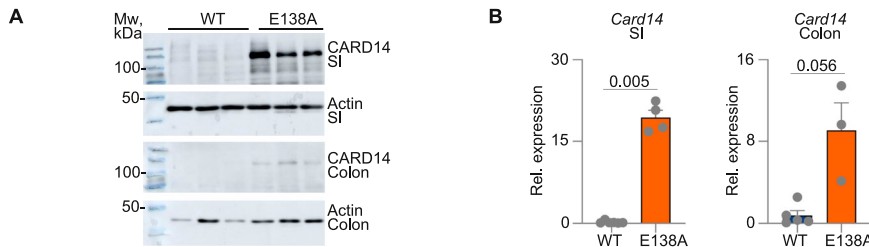

**Figure EV1.  CARD14 expression in IEC lysates.**

(**A**) Representative western blot of CARD14 in IEC lysates of small intestine and colon from control (WT) and *CARD14*(E138A)[IEC] mice. Actin was used as a loading control. Each lane represents one mouse. (**B**) Relative mRNA expression of human *CARD14* in IEC lysates from the small intestine of WT and *CARD14*(E138A)[IEC] mice. Data are presented as mean ± SEM; each symbol represents one mouse. Statistical analysis was performed using a Mann–Whitney *U* test with multiple comparisons (**B**). One representative experiment of two independent experiments is shown. Source data are available online for this figure.

A

Small intestine

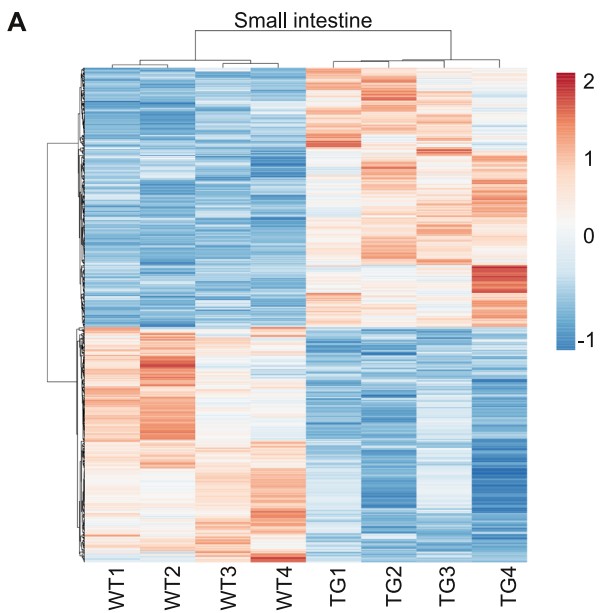

B

Colon

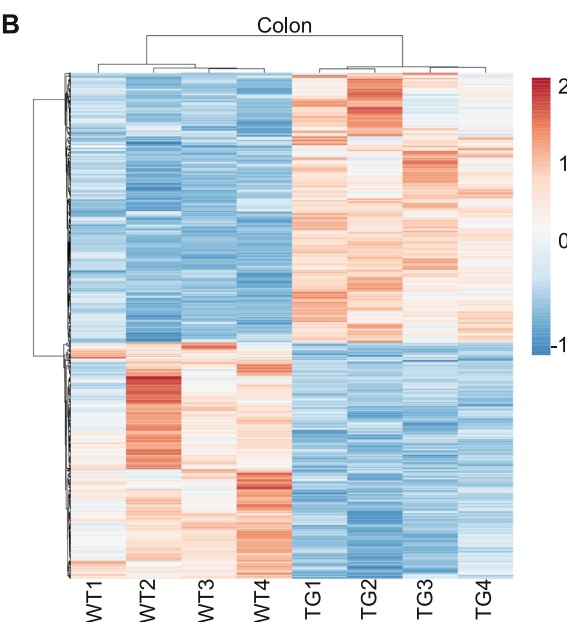

C

Gut hormones

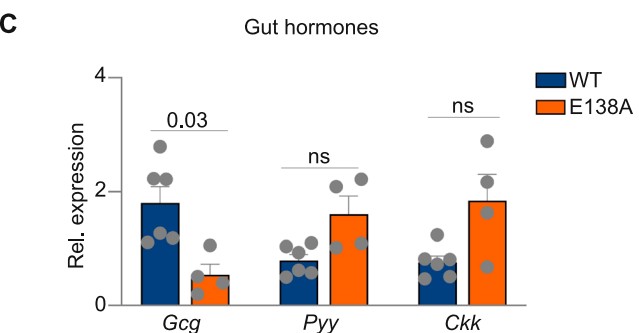

D

Bile acids markers

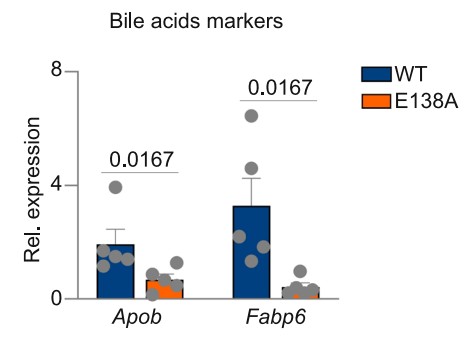

E

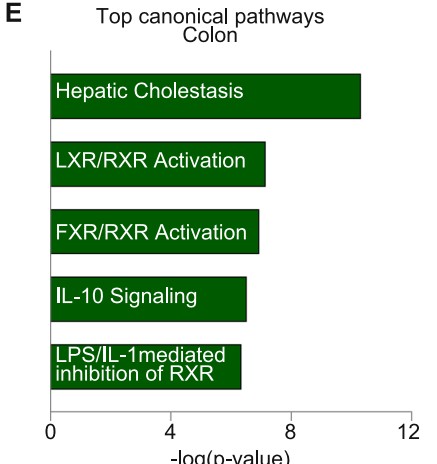

F

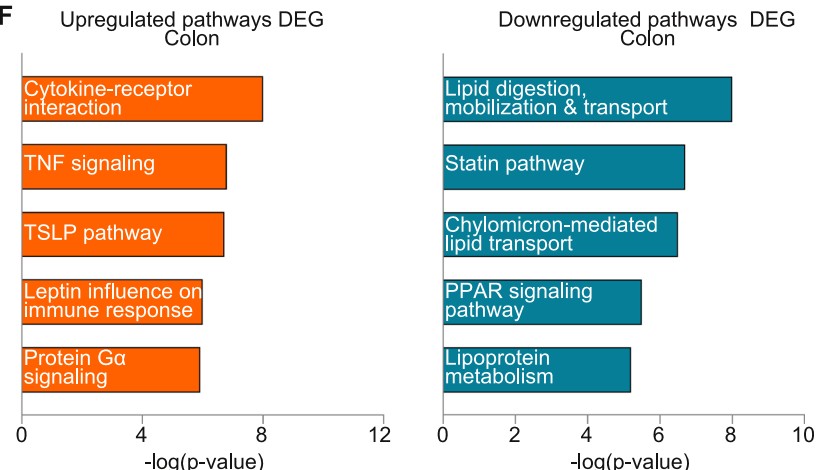

◀ **Figure EV2.  Transcriptional changes in IEC of *CARD14*(E138A)[IEC] mice.**

(A, B) Heatmaps representing differentially expressed genes (DEGs) in IEC from the small intestine (C) (1289 DEGs) and colon (D) (1626 DEGs) of WT and *CARD14*(E138A)[IEC] mice ($n = 4$, each group). (C, D) Relative mRNA expression of gut hormones (C) and bile acid metabolism-related genes (D) in colonic IEC. (E) Top canonical pathways enriched in DEGs in colon IEC from WT and *CARD14*(E138A)[IEC] mice, analyzed by Ingenuity Pathway Analysis. (F) Gene ontology enrichment analysis of biological processes for DEGs in colon IEC isolated from WT and *CARD14*(E138A)[IEC] mice, using Enrichr. Data are presented as mean ± SEM; each symbol represents one mouse. Heatmaps were generated in ClustVis from normalized counts of DEGs. Values are displayed as z-scores across each gene, no additional statistical tests were applied (A, B). Statistical analysis was performed using a Mann–Whitney *U* test with multiple comparisons (C, D), a Fisher's exact test (E, F). Data are from one independent experiment, except for (C) which includes data from two independent experiments. Source data are available online for this figure.

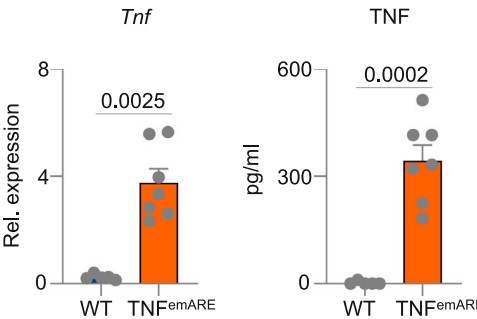

**Figure EV3.   Relative mRNA expression and protein levels of TNF of ileal lysates from control (WT) and TNF^emARE mice.**

Data are presented as mean ± SEM; each symbol represents one mouse. Statistical analysis was performed using a Mann–Whitney *t* test (relative *Tnf* gene expression) or unpaired *t* test with Welch correction (TNF protein level). One representative experiment of two independent experiments is shown. Source data are available online for this figure.

