## [Peer Review File · EMBO Molecular Medicine]

CARD14 signaling in intestinal epithelial cells induces intestinal inflammation and intestinal transit delay

Aigerim Aidarova, Marieke Carels, Mira Haegman, Yasmine Driège, Steven Timmermans, Eline Van Damme, Javier Aguilera-Lizarraga, Maria Viola, Rita de Cássia Collaço, Joan Manils, Steven Ley, Frank Bosmans, Tom Van de Wiele, Guy Boeckxstaens, Claude Libert, Rudi Beyaert, and Inna Afonina

Corresponding authors: Inna Afonina (inna.afonina@irc.vib-ugent.be) , Rudi Beyaert (Rudi.Beyaert@irc.vib-ugent.be)

Review Timeline:

Submission Date:	10th Apr 25
Editorial Decision:	30th Apr 25
Revision Received:	8th Aug 25
Editorial Decision:	15th Sep 25
Revision Received:	22nd Sep 25
Accepted:	29th Sep 25

Editor: Lise Roth

Transaction Report:

30th Apr 2025

Dear Dr. Afonina,

Thank you for the submission of your manuscript to EMBO Molecular Medicine. We have now received feedback from the three reviewers who agreed to evaluate your manuscript. As you will see from the reports below, the referees acknowledge the interest of the study, but also request more in-depth mechanistic understanding. Referee #3 further comments on the lack of human relevance, which is a critical point for consideration in EMBO Molecular Medicine.

If you feel you can satisfactorily address these concerns and those listed by the referees, you may wish to submit a revised version of your manuscript. Please attach a covering letter giving details of the way in which you have handled each of the points raised by the referees. A revised manuscript will once again be subject to review and we cannot guarantee at this stage that the eventual outcome will be favorable.

Addressing the reviewers' concerns in full will be necessary for further considering the manuscript in our journal, and acceptance of the manuscript will entail a second round of review. EMBO Molecular Medicine encourages a single round of revision only and therefore, acceptance or rejection of the manuscript will depend on the completeness of your responses included in the next, final version of the manuscript. For this reason, and to save you frustration at the end, I would strongly discourage you from returning an incomplete revision.

We are expecting your revised manuscript within three to four months, if you anticipate any delay, please contact us.

We require:

- 1) A .docx formatted version of the manuscript text (including legends for main figures, EV figures and tables). Please make sure that the changes are highlighted to be clearly visible.
- 2) Individual production quality figure files as .eps, .tif, .jpg (one file per figure). For guidance, download the 'Figure Guide PDF' (<https://www.embopress.org/page/journal/17574684/authorguide#figureformat>).
- 3) At EMBO Press we ask authors to provide source data for the main figures. Our source data coordinator will contact you to discuss which figure panels we would need source data for and will also provide you with helpful tips on how to upload and organize the files.
- 4) A .docx formatted letter INCLUDING the reviewers' reports and your detailed point-by-point responses to their comments. As part of the EMBO Press transparent editorial process, the point-by-point response is part of the Review Process File (RPF), which will be published alongside your paper.
- 5) A complete author checklist, which you can download from our author guidelines (<https://www.embopress.org/page/journal/17574684/authorguide#submissionofrevisions>). Please insert information in the checklist that is also reflected in the manuscript. The completed author checklist will also be part of the RPF.
- 6) All Materials and Methods need to be described in the main text using our 'Structured Methods' format. According to this format, the Methods section includes a Reagents and Tools Table (listing key reagents, experimental models, software and relevant equipment and including their sources and relevant identifiers) followed by a Methods and Protocols section describing the methods, ideally using a step-by-step protocol format. The aim is to facilitate adoption of the methodologies across labs. Please download and fill our Reagents and Tools Table template (.docx), which you can find in our author guidelines: <https://www.embopress.org/page/journal/14693178/authorguide#structuredmethods>. When submitting your revised manuscript, please do not include the Reagents and Tools Table in the Methods section of the manuscript but upload it as a separate file choosing the file type "Reagent Table".
- 7) Please note that all corresponding authors are required to supply an ORCID ID for their name upon submission of a revised manuscript.

8) It is mandatory to include a 'Data Availability' section after the Materials and Methods. Before submitting your revision, primary datasets produced in this study need to be deposited in an appropriate public database, and the accession numbers and database listed under 'Data Availability'. Please remember to provide a reviewer password if the datasets are not yet public (see <https://www.embopress.org/page/journal/17574684/authorguide#dataavailability>).

9) For data quantification: please specify the name of the statistical test used to generate error bars and P values, the number (n) of independent experiments (specify technical or biological replicates) underlying each data point and the test used to calculate p-values in each figure legend. The figure legends should contain a basic description of n, P and the test applied. Graphs must include a description of the bars and the error bars (s.d., s.e.m.). Please provide exact p values.

10) Our journal encourages inclusion of *data citations in the reference list* to directly cite datasets that were re-used and obtained from public databases. Data citations in the article text are distinct from normal bibliographical citations and should directly link to the database records from which the data can be accessed. In the main text, data citations are formatted as follows: "Data ref: Smith et al, 2001" or "Data ref: NCBI Sequence Read Archive PRJNA342805, 2017". In the Reference list, data citations must be labeled with "[DATASET]". A data reference must provide the database name, accession number/identifiers and a resolvable link to the landing page from which the data can be accessed at the end of the reference. Further instructions are available at .

11) We replaced Supplementary Information with Expanded View (EV) Figures and Tables that are collapsible/expandable online. EV Figures should be cited as 'Figure EV1, Figure EV2' etc... in the text and their respective legends should be included in the main text after the legends of regular figures.

12) The paper explained: EMBO Molecular Medicine articles are accompanied by a summary of the articles to emphasize the major findings in the paper and their medical implications for the non-specialist reader. Please provide a draft summary of your article highlighting

13) Author contributions: CRedit has replaced the traditional author contributions section because it offers a systematic machine readable author contributions format that allows for more effective research assessment. Please remove the Authors Contributions from the manuscript and use the free text boxes beneath each contributing author's name in our system to add specific details on the author's contribution. More information is available in our guide to authors.

Please also suggest a visual abstract to illustrate your article as a PNG file 550 px wide x 300-600 px high. A cropped portion of this image will serve as thumbnail for the table of content on our webpage.

16) As part of the EMBO Publications transparent editorial process initiative (see our Editorial at <http://embomolmed.embopress.org/content/2/9/329>), EMBO Molecular Medicine will publish online a Review Process File (RPF) to accompany accepted manuscripts.

In the event of acceptance, this file will be published in conjunction with your paper and will include the anonymous referee reports, your point-by-point response and all pertinent correspondence relating to the manuscript. Let us know whether you agree with the publication of the RPF and as here, if you want to remove or not any figures from it prior to publication. Please note that the Authors checklist will be published at the end of the RPF.

I look forward to receiving your revised manuscript.

Yours sincerely,

Lise Roth

***** Reviewer's comments *****

Referee #1 (Comments on Novelty/Model System for Author):

The authors present an elegant and comprehensive study elucidating the effects of gain-of-function CARD14 signaling in intestinal epithelial cells (IECs). By generating novel transgenic mouse models, they uncover an unexpected link between CARD14 activation, mild intestinal inflammation, impaired intestinal transit, Paneth cell dysfunction, microbial dysbiosis, and increased susceptibility to enteric infection. The manuscript is well-structured, methodologically sound, and provides significant new insights into the previously unexplored role of CARD14 in intestinal biology. The experiments are thorough, the findings are intriguing, and the paper will be of broad interest to researchers in immunology, gastroenterology, and mucosal biology.

Referee #1 (Remarks for Author):

The authors present an elegant and comprehensive study elucidating the effects of gain-of-function CARD14 signaling in intestinal epithelial cells (IECs). By generating novel transgenic mouse models, they uncover an unexpected link between CARD14 activation, mild intestinal inflammation, impaired intestinal transit, Paneth cell dysfunction, microbial dysbiosis, and increased susceptibility to enteric infection. The manuscript is well-structured, methodologically sound, and provides significant new insights into the previously unexplored role of CARD14 in intestinal biology. The experiments are thorough, the findings are intriguing, and the paper will be of broad interest to researchers in immunology, gastroenterology, and mucosal biology. However, there are a few points that need clarification, expansion, or improvement before publication.

- i) While the data convincingly show that CARD14 activation delays intestinal transit, the proposed mechanisms remain somewhat speculative. The manuscript would benefit from a more mechanistic exploration (or discussion) of how epithelial changes translate into altered neuronal function or motility without observable neuronal death or dysfunction;
- ii) The results indicate that antibiotic treatment abolishes the additional transit delay caused by CARD14 activation. However, antibiotics themselves slow transit even in wild-type mice, complicating the interpretation;
- iii) The susceptibility to *C. rodentium* infection is an important and well-conducted experiment. However, it would be helpful to explicitly state whether the increased bacterial burden and weight loss are due solely to reduced AMP production, or whether impaired motility could also contribute by prolonging bacterial contact with the mucosa.

Minor points:

Some of the figure legends are quite dense. Consider simplifying them for clarity, without losing essential details; For some experiments with relatively small sample sizes (e.g., n=3-5), ensure that the choice of statistical test is always appropriate and conservative.

Referee #2 (Comments on Novelty/Model System for Author):

These findings are highly interesting for our understanding of the role of CARD14 in promoting inflammatory conditions that go beyond skin manifestations. Also, the mouse model of mild intestinal inflammation that the authors developed should be an asset for future studies.

Overall, the data are throughout of high technical quality.
I have only a few concerns/suggestions for improvement.

Referee #2 (Remarks for Author):

The manuscript by Aidarova et al. addresses the effects of an intestinal overexpression of an active mutant of CARD14 in intestinal epithelial cells of mice. The authors observe development of mild intestinal inflammation that correlated with intestinal infiltration of neutrophils, eosinophils and dendritic cells, and upregulation of inflammatory mediators in IECs, which was confirmed by transcriptome analysis. Mice also showed reduced intestinal motility, which depended on MALT1 scaffold but not protease function. Enteric neuronal survival and function was unaffected, but functional changes were observed in intestinal secretory functions, in particular a strong decrease in AMP secretion and downregulation of Paneth cell markers, which was not due to a cell intrinsic effect on Paneth cells. Finally, an analysis of the gut microbiome revealed a significant reduction in microbial diversity, which likely contributed to decreased intestinal motility, and an increased susceptibility to intestinal infection by *Citrobacter Rodentium*.

These findings are highly interesting for our understanding of the role of CARD14 in promoting inflammatory conditions that go beyond skin manifestations. Also, the mouse model of mild intestinal inflammation that the authors developed should be an asset for future studies.

Overall, the data are throughout of high technical quality.

I have only a few concerns/suggestions for improvement.

Major concerns:

1) It would be interesting to see to which extent the protease activity of MALT1 might still contribute to the inflammation effect of CARD14(E138A), since 2 different mechanisms may contribute to regulate those processes: would it be possible to perform experiments 3B and 3C using not only the MALT1 KO mice but also mice treated with the MALT1 protease inhibitor? Or to check this at least on isolated IECs treated with or without the MALT1 inhibitor?

2) Cck and Pyy are examples of genes identified as upregulated, but other genes of interest cannot be identified from the Volcano plots provided in Fig. 5A and 5B. The full data of the RNASeq analysis should be made available in the supplementary data or on a publicly accessible server.

Minor points:

1) A recent publication that may have escaped the authors' attention (Lundquist et al. BBA - Mol Basis of Disease 2025) establishes a correlation between psoriasis and subclinical intestinal disease. These findings should be discussed and set in relation to the authors findings here.

2) In the results text for figure 3, there is a typo: CARD14(138A) instead of CARD14(E138A).

3) In a few parts of the text the " " in NF- B is missing.

4) I may have missed it: in figure 1E and other qPCR data, which gene was used to normalize the data? And why did the authors use different statistical tests for the different genes analyzed?

5) The conclusion of results of the enteric neuronal section that reduced intestinal motility in CARD14(E138A)IEC mice is caused by IEC-intrinsic changes "that affect epithelial-neuronal communications in the intestine", seems to be an overstatement, the authors may want to rephrase that sentence.

Referee #3 (Comments on Novelty/Model System for Author):

The findings in this mouse model is not reproducible in human patients. The authors did not provide the mechanistic insight into how the gain-of-function CARD14 mutation in IEC caused a variety of phenotypes.

Referee #3 (Remarks for Author):

In this manuscript, the authors generated mice expressing human gain-of-function CARD14 (E138A) in intestinal epithelial cells (IEC) and analyzed their phenotypes. CARD14 (E138A)IEC mice showed several phenotypes, including low grade intestinal inflammation, reduced intestinal motility, impaired neuron-evoked anion secretion in the intestine, Paneth cell dysfunction, and dysbiosis.

The findings in this study are interesting; however, the manuscript is descriptive. In the present form, the authors just show several phenotypes in the mutant mice. The authors should clearly show the mechanism how the gain-of-function of CARD14 mutation in IEC leads to the appearance of those phenotypes.

The reviewer imagines that the gain-of-function of CARD14 mutation in IEC leads to MALT1-dependent activation of NF- κ B and MAP kinase and thereby induce expression of pro-inflammatory genes and chemokine genes. This develops mild intestinal inflammation and the recruitment of inflammatory cells including neutrophils. Based on the literature, TNF might induce dysfunction of Paneth cells. The reviewer wonders if this logic is applicable, but the authors should show some mechanistic insight into how the gain-of-function of CARD14 mutation in IEC leads to development of variety of phenotypes. If this is the case (inflammation is induced first by the gain-of-function of CARD14 mutation in IEC), the findings are predictable.

Specific points:

1. The authors should show Paneth cell dysfunction is induced by TNF by introducing TNF deficiency into CARD14 (E138A)IEC mice.
2. The authors should also show whether the inflammation causes the delayed intestinal motility. In Fig.3, the authors show that MLT827 did not recover the intestinal motility in the mutant mice. In this case, was intestinal inflammation not cancelled?
3. The authors should also show the mechanisms into how epithelial-neuronal communication is induced. Is this inflammation-dependent?
4. In the final part of Discussion, the authors stated that patients with the gain-of-function of CARD14 mutation do not show intestinal inflammation. This reduces the value of this study.

Reply to the editor (Dr. Lise Roth):

We were happy to read in your e-mail of May 20th that you and your colleagues agreed with our proposed revisions, including new data and discussion, which we shared with you via e-mail on May 13th.

We here provide you with a revised manuscript that has been changed accordingly. Below, we also provide detailed point-by-point responses to the reviewers' comments. In addition, we have reviewed and followed all the journal instructions that were outlined in your e-mail of May 13th.

Point-by-point response to the reviewers' comments:

We thank the Reviewers for their thoughtful comments, which allowed us to further improve our manuscript. All revisions are marked in the revised draft. We hope the updated draft is now acceptable for publication in EMBO Molecular Medicine.

REFEREE #1:

The authors present an elegant and comprehensive study elucidating the effects of gain-of-function CARD14 signaling in intestinal epithelial cells (IECs). By generating novel transgenic mouse models, they uncover an unexpected link between CARD14 activation, mild intestinal inflammation, impaired intestinal transit, Paneth cell dysfunction, microbial dysbiosis, and increased susceptibility to enteric infection. The manuscript is well-structured, methodologically sound, and provides significant new insights into the previously unexplored role of CARD14 in intestinal biology. The experiments are thorough, the findings are intriguing, and the paper will be of broad interest to researchers in immunology, gastroenterology, and mucosal biology.

RESPONSE: we are glad to read that the reviewer finds our manuscript of high quality and of broad interest.

However, there are a few points that need clarification, expansion, or improvement before publication.

i) While the data convincingly show that CARD14 activation delays intestinal transit, the proposed mechanisms remain somewhat speculative. The manuscript would benefit from a more mechanistic exploration (or discussion) of how epithelial changes translate into altered neuronal function or motility without observable neuronal death or dysfunction;

RESPONSE: We agree with the reviewer that the underlying mechanism for the observed change in intestinal motility is still somewhat speculative. Given the complex pathophysiology of intestinal motility and broad range of processes and cell types involved, in many cases acting bidirectional, elucidating the full mechanism is very challenging. Although we show a strong decrease in the production of several antimicrobial peptides, which are produced by Paneth cells, our original data already allowed us to exclude a role for Paneth cell-intrinsic CARD14(E138A) signaling and indicate a role for CARD14(E138A) signaling in other types of IEC (see also below). Neutrophil depletion experiments also excluded a role for CARD14(E138A)-induced neutrophilia in the intestine.

Inflammatory cytokines such as TNF, whose expression is upregulated in the IEC of CARD14(E138A)-expressing mice, were previously shown to induce Paneth cell dysfunction. Interestingly, increased TNF expression has also been associated with reduced fecal output and water content in aged mice, which was reversed by administration of a TNF antagonist. However, treatment with TNF neutralizing

antibodies did not prevent the CARD14(E138A)-induced delay in intestinal transit. The above has now been added in the results (page 10) and discussion (page 13) section of the revised manuscript and is shown as Fig. 6J. In addition, we now also analyzed intestinal transit time in TNF^{emARE} mice, which overexpress TNF and spontaneously develop intestinal inflammation. TNF^{emARE} mice and WT control mice showed similar intestinal motility (Fig. 6K in the revised manuscript). Together, these data indicate that TNF is not a central driver of delayed intestinal motility in CARD14(E138A)^{IEC} mice, and that other mechanisms likely contribute to Paneth cell dysfunction and impaired motility.

As we also demonstrate that neuromuscular function itself remains intact in CARD14(E138A)^{IEC} mice, inflammation-induced neuronal loss can also be excluded. Although we cannot fully eliminate a causal role for inflammation in the decrease of intestinal motility in our mouse model, the above observations make this rather unlikely.

Following the reviewer's request, we now discuss in the revised manuscript how epithelial changes may translate into altered neuronal function or motility without observable neuronal death or dysfunction, and illustrate this with extra data obtained from the RNA seq analysis. More specifically, our RNAseq data provide evidence for changes in gut hormones, which are produced by enteroendocrine cells (EEC) in the intestine, and genes related to bile acid metabolism (decreased *Abcg5*, *ApoB*, *Fabp6*, and increased *Fgf15* in colon; revised Fig. 5B). In the revised manuscript, both were also validated by qPCR (Fig. EV2C-D). Differentially expressed genes in CARD14(E138A)^{IEC} and WT control mice were further analysed using Ingenuity Pathway Analysis, which confirmed the activation of canonical bile acid-related pathways such as LXR/RXR and FXR/RXR signaling (Fig. EV2E of the revised manuscript). Moreover, gene ontology pathway analysis in colon confirmed the activation of inflammatory signaling and also indicated the downregulation of genes associated with lipid metabolism and transport, which may be functionally linked to alterations in bile acid metabolism (Fig. EV2F of the revised manuscript). The above are now mentioned on page 8 in the results section of the revised manuscript. In addition, the text below has been added in the discussion section of the revised manuscript (pages 13-14):

"Instead our data support more a role for impaired epithelial-neuronal communication, in which also changes in microbiota can be implicated. In this context, RNAseq of IEC from CARD14(E138A)^{IEC} mice indicated a potential role for several gut hormones (Cck, Pyy, Gcg) that are produced by specialized intestinal EEC in response to various stimuli, including microbial metabolites or metabolites derived from microbial fermentation, such as short-chain fatty acids. Gut hormones, in turn, regulate numerous functions, including gastric emptying and intestinal motility. For example, cholecystokinin (CCK) is known to mediate sensory and motor responses to intestinal distension and is believed to contribute to the symptoms of constipation, bloating and abdominal pain, while CCK receptor antagonists have been tested in patients with constipation. Peptide YY (PYY) was shown before to increase intestinal water and electrolyte absorption and exhibit antisecretory activity, which could explain the reduced neuron-evoked ion secretion in CARD14(E138A)^{IEC} mice. Interestingly, PYY-cell density is also increased in patients with irritable bowel syndrome with constipation. Gut hormones further act through specific receptors on enteric neurons, modulating gut motility through sensory and motor pathways. It still remains to be investigated whether EEC-intrinsic CARD14 signaling is responsible for the dysregulated expression of gut hormones, or whether the observed changes are caused by changes in gut microbiota directly influencing EEC function, maturation, and hormone secretion.

Interestingly, RNAseq data of IEC isolated from CARD14(E138A)^{IEC} mice and Ingenuity Pathway Analysis also hints to changes in bile acid metabolism. After their synthesis from cholesterol in the hepatocyte, bile acids pass into the small intestine where they act as detergents to emulsify fats, aiding in their digestion and absorption, and also function as signaling molecules that activate various nuclear and G protein-coupled receptors to regulate multiple cellular processes and functions in the intestine and other tissues. Certain bile acids are known to have a laxative effect, while other studies suggest that decreased fecal conjugated bile acids are associated with constipation. The size and composition of the bile acid pool is strongly regulated by their efficient enterohepatic

recirculation, metabolism, and the homeostatic feedback mechanisms connecting hepatocytes and enterocytes. Importantly, also the luminal gut microbiome is known to modulate bile acid composition via deconjugation of the primary bile acids. Changes in the intestinal microbiome of CARD14(E138A)^{IEC} mice could therefore affect epithelial-neuronal communication by altering the gut bile acid composition. Given the complexity and redundancy of gut hormone and bile acid signaling, and their bidirectional relationship with the gut microbiota, it will be very difficult to pinpoint the specific role of each in the regulation of intestinal motility in CARD14(E138A)^{IEC} mice.”

ii) The results indicate that antibiotic treatment abolishes the additional transit delay caused by CARD14 activation. However, antibiotics themselves slow transit even in wild-type mice, complicating the interpretation;

RESPONSE: Indeed, antibiotic treatment also has an effect in wild type mice, which we also acknowledged in the original version of the manuscript.

To further investigate the role of the intestinal microbiome we now also analysed intestinal motility in our mouse model under germ-free conditions (Fig. 7C of the revised manuscript), which further supported the role of the microbiome.

The following text was added in the results section of the revised manuscript (page 11)

“To further validate the role of the microbiome, we rederived CARD14(E138A)^{IEC} and control WT mice under germ-free conditions in the germ-free and gnotobiotic mouse core facility of our center. In agreement with the results after antibiotics treatment, no significant differences in gastrointestinal transit time were observed between CARD14(E138A)^{IEC} and control WT mice in the absence of microbiome (Fig. 7C). These data further support the role of the microbiome in CARD14(E138A)-induced intestinal dysmotility.”

However, and in agreement with previous reports demonstrating a key regulatory role of the microbiome in intestinal motility, both antibiotics or germ-free conditions themselves slow transit in wild-type mice, complicating somewhat the interpretation of our results. This was also added to the discussion section (page 13 of the revised manuscript). As CARD14(E138A) did not further delay intestinal transit compared to WT mice under both conditions, we are quite confident to conclude on a role of the microbiome in our model.

iii) The susceptibility to C. rodentium infection is an important and well-conducted experiment. However, it would be helpful to explicitly state whether the increased bacterial burden and weight loss are due solely to reduced AMP production, or whether impaired motility could also contribute by prolonging bacterial contact with the mucosa.

RESPONSE: We thank the reviewer for pointing this out. The following passage has been added to the discussion (page 13) to acknowledge different possibilities:

“However, we cannot exclude that increased sensitivity also partially results from the longer transit time in CARD14(E138A)^{IEC} mice, prolonging bacterial contact with the mucosa.”

Minor points:

Some of the figure legends are quite dense. Consider simplifying them for clarity, without losing essential details;

RESPONSE: Figure legends have been adjusted.

For some experiments with relatively small sample sizes (e.g., n=3-5), ensure that the choice of statistical test is always appropriate and conservative.

RESPONSE: Statistical analysis was performed in consultation with a statistician (Marnik Vuylsteke) who is providing his expertise to our lab as a consultant. Appropriate statistical test was chosen for each experiment based on the type of data and experimental set up. Data was re-analysed and specific details are given in materials and methods or figure legends.

REFEREE #2:

The manuscript by Aidarova et al. addresses the effects of an intestinal overexpression of an active mutant of CARD14 in intestinal epithelial cells of mice. The authors observe development of mild intestinal inflammation that correlated with intestinal infiltration of neutrophils, eosinophils and dendritic cells, and upregulation of inflammatory mediators in IECs, which was confirmed by transcriptome analysis. Mice also showed reduced intestinal motility, which depended on MALT1 scaffold but not protease function. Enteric neuronal survival and function was unaffected, but functional changes were observed in intestinal secretory functions, in particular a strong decrease in AMP secretion and downregulation of Paneth cell markers, which was not due to a cell intrinsic effect on Paneth cells. Finally, an analysis of the gut microbiome revealed a significant reduction in microbial diversity, which likely contributed to decreased intestinal motility, and an increased susceptibility to intestinal infection by *Citrobacter Rodentium*.

These findings are highly interesting for our understanding of the role of CARD14 in promoting inflammatory conditions that go beyond skin manifestations. Also, the mouse model of mild intestinal inflammation that the authors developed should be an asset for future studies.

Overall, the data are throughout of high technical quality.

RESPONSE: we thanks the reviewer for the very positive comments.

I have only a few concerns/suggestions for improvement.

Major concerns:

1) It would be interesting to see to which extent the protease activity of MALT1 might still contribute to the inflammation effect of CARD14(E138A), since 2 different mechanisms may contribute to regulate those processes: would it be possible to perform experiments 3B and 3C using not only the MALT1 KO mice but also mice treated with the MALT1 protease inhibitor? Or to check this at least on isolated IECs treated with or without the MALT1 inhibitor?

RESPONSE: We did not analyze the effect of MALT1 inhibitor treatment on CARD14(E138A)-induced neutrophilia in our original experiments shown in the manuscript and did not get ethical approval for another experiment. However, we have now also analysed proinflammatory gene expression in IEC from control and CARD14(E138A)^{IEC} mice treated or not with the MALT1 inhibitor in the original experiments. Although, MALT1 inhibitor treatment clearly suppressed MALT1 activity as evidenced by the inhibition of the CARD14(E138A)-induced CYLD cleavage (figure 3D), CARD14(E138A)-induced expression of Tnf and Cxcl2 were unaffected (new Figure 3E). These data suggest that MALT1 proteolytic activity does not have a major contribution to the inflammatory effect of CARD14(E138A) in the intestine. It should be mentioned that MALT1 catalytic activity is known to only affect the expression of a limited number of genes in activated T cells, most likely reflecting the cleavage of RNA destabilizing proteins such as Roquin. We can therefore not fully exclude a similar effect of MALT1 catalytic activity on the expression of other genes in IEC.

2) Cck and Pyy are examples of genes identified as upregulated, but other genes of interest cannot be identified from the Volcano plots provided in Fig. 5A and 5B. The full data of the RNASeq analysis should be made available in the supplementary data or on a publicly accessible server.

RESPONSE: The raw RNA sequencing data were uploaded on Gene Expression Omnibus (GEO; <https://www.ncbi.nlm.nih.gov/geo/>) under accession number GSE302705, a public functional genomics data repository, and will become available upon publishing of the manuscript. We have now also added genes related to bile acid metabolism in the Volcano plot in Fig. 5B. A comprehensive overview of all the upregulated and downregulated genes in small intestine and colon is now also listed in the Suppl. Table EV1 and EV2, respectively, of the revised manuscript.

Minor points:

1) A recent publication that may have escaped the authors' attention (Lundquist et al. BBA - Mol Basis of Disease 2025) establishes a correlation between psoriasis and subclinical intestinal disease. These findings should be discussed and set in relation to the authors findings here.

We thank the reviewer for bringing this highly relevant paper to our attention, which is now also mentioned in the discussion section. In addition, we refer now also to two other studies (PMID: 9217818 and PMID: 7066192) that support a link between psoriasis and intestinal disease.

2) In the results text for figure 3, there is a typo: CARD14(138A) instead of CARD14(E138A).

RESPONSE: This has been corrected.

3) In a few parts of the text the "κ" in NF-κB is missing.

RESPONSE: This has been corrected.

4) I may have missed it: in figure 1E and other qPCR data, which gene was used to normalize the data? And why did the authors use different statistical tests for the different genes analyzed?

RESPONSE: As a standard procedure we are including up to 4 housekeeping genes (Actb, Hprt1, Ubc, Rpl13a) in each run. During analysis, geNorm algorithm is used to determine the most stable housekeeping genes, as well as the optimal number of reference genes needed for accurate normalization (usually 2-3). The actual housekeeping genes used in each run may therefore be different between the experiments.

We have further consulted the statistician Marnik Vuylsteke, who is providing his expertise to our lab as a consultant. We have reanalysed the data for qPCR using Mann-Whitney U-test, with P value corrected for multiple testing. Statistical analysis between four groups was done using a log linear regression model. Specific details are given in materials and methods or figure legends.

5) The conclusion of results of the enteric neuronal section that reduced intestinal motility in CARD14(E138A)IEC mice is caused by IEC-intrinsic changes "that affect epithelial-neuronal communications in the intestine", seems to be an overstatement, the authors may want to rephrase that sentence.

RESPONSE: Thank you for pointing this out. The sentence has been rephrased to tone down our conclusion: *'Together, these data demonstrate that the survival and cell-intrinsic functionality of enteric neurons is unaffected in ileum of CARD14(E138A)^{IEC} mice, and indicate that reduced intestinal motility in CARD14(E138A)^{IEC} mice may result from IEC-intrinsic changes that affect epithelial-neuronal communication in the intestine, rather than from enteric neuronal loss or dysfunction.'*

REFEREE #3:

In this manuscript, the authors generated mice expressing human gain-of-function CARD14 (E138A) in intestinal epithelial cells (IEC) and analyzed their phenotypes. CARD14 (E138A)^{IEC} mice showed several phenotypes, including low grade intestinal inflammation, reduced intestinal motility, impaired neuron-evoked anion secretion in the intestine, Paneth cell dysfunction, and dysbiosis.

The findings in this study are interesting; however, the manuscript is descriptive. In the present form, the authors just show several phenotypes in the mutant mice. The authors should clearly show the mechanism how the gain-of-function of CARD14 mutation in IEC leads to the appearance of those phenotypes.

RESPONSE: Given the complex pathophysiology of intestinal motility and broad range of processes and cell types involved, in many cases acting bidirectional, elucidating the full mechanism is very challenging. However, we respectfully disagree with the conclusion that our manuscript is purely descriptive and hope we can convince the reviewer based on the following evidence and extra experiments we performed:

The phenotypes we describe in our mice are: mild intestinal inflammation, decreased intestinal motility, and increased sensitivity to enteric infection. Although, we do not yet fully understand the underlying mechanisms and the relation between each of these phenotypes, we do provide some clear insights for each of them. 1. Mechanisms underlying mild intestinal inflammation: we show that hyperactive CARD14(E138A) signaling in IECs induces the expression of several proinflammatory cytokines and chemokines that are known to mediate intestinal neutrophil infiltration and inflammation. Increased proinflammatory gene expression most likely reflects the NF- κ B activating potential of CARD14 that is mediated via the MALT1 scaffold function (as shown by the MALT1 knockout data); 2. Mechanisms underlying decreased intestinal dysmotility: we show that hyperactive CARD14(E138A) signaling in IECs decreases the production of antimicrobial peptides by Paneth cells and induces intestinal dysbiosis, which is known to contribute to constipation in humans. A role for intestinal dysbiosis is also supported by both our antibiotic treatment experiment and the germ-free mouse experiment (now added to the revised manuscript). Moreover, we also show that CARD14(E138A) alters gut hormone expression by enteroendocrine cells and bile acid metabolism (added to the revised manuscript), which is known to affect intestinal motility; 3. Mechanisms underlying increased intestinal Citrobacter infection: as Paneth cells are critically important to control intestinal bacterial infection, the observed Paneth cell dysfunction is also most likely contributing to increased Citrobacter infection in CARD14(E138A)^{IEC} mice. However, as mentioned by reviewer 1, we cannot exclude that increased sensitivity also reflects a longer bacterial exposure of the gut due to the longer transit time (this has now also been added in the revised manuscript).

The reviewer imagines that the gain-of-function of CARD14 mutation in IEC leads to MALT1-dependent activation of NF- κ B and MAP kinase and thereby induce expression of pro-inflammatory genes and chemokine genes. This develops mild intestinal inflammation and the recruitment of inflammatory cells including neutrophils. Based on the literature, TNF might induce dysfunction of Paneth cells. The reviewer wonders if this logic is applicable, but the authors should show some mechanistic insight into how the gain-of-function of CARD14 mutation in IEC leads to

development of variety of phenotypes. If this is the case (inflammation is induced first by the gain-of-function of CARD14 mutation in IEC), the findings are predictable.

RESPONSE: We don't believe that the above logic is fully applicable to our findings. As shown in our original manuscript, we already excluded a role of neutrophils in decreased intestinal motility in our model. To study the role of CARD14-induced NF- κ B signaling, we also analyzed the effect of IKK inhibitor treatment, but found that the latter was associated with severe toxicity in CARD14(E138A)^{IEC} mice (in agreement with the known cell death protective effect of NF- κ B in the intestine), preventing us from measuring intestinal motility (we therefore decided not to include these data in the main manuscript but provide the corresponding figure 1 for the information of the reviewer in this rebuttal). Based on literature, a possible role for TNF can indeed be proposed and we further analyzed this using two alternative approaches (see reply to reviewer 1 and below as reply to specific point 1).

The exact mechanism by which increased CARD14 signaling in IEC results in Paneth cell dysfunction remains unclear, but we could exclude a Paneth cell-intrinsic effect of CARD14 signaling. We have also demonstrated that neuromuscular function itself remains intact in CARD14(E138A)^{IEC} mice, further excluding inflammation-induced neuronal loss. Our findings suggest a role for impaired epithelial-neuronal communication in our mouse model and extra data obtained from the RNA seq analysis indicated several interesting candidates (revised Fig. 5B; Suppl.

Fig. EV2C-D). However, because most of these messengers (e.g. gut hormones, bile acids, ...) are known to act in concert to affect intestinal motility and also have multiple other activities, it will be very difficult to pinpoint the specific role of one specific molecule. We therefore favor the suggestion of reviewer 1 to expand our discussion section with a more mechanistic exploration of how epithelial changes translate into altered neuronal function or motility without observable neuronal death or dysfunction. The text below (also provided in our response to point (i) of referee 1) has been added in the discussion section of the revised manuscript (pages 13-14):

"Instead our data support more a role for impaired epithelial-neuronal communication, in which also changes in microbiota can be implicated. In this context, RNAseq of IEC from CARD14(E138A)^{IEC} mice indicated a potential role for several gut hormones (Cck, Pyy, Gcg) that are produced by specialized intestinal EEC in response to various stimuli, including microbial metabolites or metabolites derived from microbial fermentation, such as short-chain fatty acids. Gut hormones, in turn, regulate numerous functions, including gastric emptying and intestinal motility. For example, cholecystokinin (CCK) is known to mediate sensory and motor responses to intestinal distension and is believed to contribute to the symptoms of constipation, bloating and abdominal pain, while CCK receptor antagonists have been tested in patients with constipation. Peptide YY (PYY) was shown before to increase intestinal water and electrolyte absorption and exhibit antisecretory activity, which could explain the reduced neuron-evoked ion secretion in CARD14(E138A)^{IEC} mice. Interestingly, PYY-cell density is also increased in patients with irritable bowel syndrome with constipation. Gut hormones further act through specific receptors on enteric neurons, modulating gut motility through sensory and motor pathways. It still remains to be investigated whether EEC-intrinsic CARD14 signaling is responsible for the dysregulated expression of gut hormones, or whether the observed changes are caused by changes in gut microbiota directly influencing EEC function, maturation, and hormone secretion.

Interestingly, RNAseq data of IEC isolated from CARD14(E138A)^{IEC} mice and Ingenuity Pathway Analysis also hints to changes in bile acid metabolism. After their synthesis from cholesterol in the

Figure 1 Percentage of body weight loss in WT control and iCARD14(E138A)^{IEC} mice following treatment with IKK inhibitor (BMS-345541, MedChemExpress, HY-10518) for 7 days. Mice received 100 mg/kg of BMS-345541 or vehicle i.p. once every day. Three mice per group were used.

hepatocyte, bile acids pass into the small intestine where they act as detergents to emulsify fats, aiding in their digestion and absorption, and also function as signaling molecules that activate various nuclear and G protein-coupled receptors to regulate multiple cellular processes and functions in the intestine and other tissues. Certain bile acids are known to have a laxative effect, while other studies suggest that decreased fecal conjugated bile acids are associated with constipation. The size and composition of the bile acid pool is strongly regulated by their efficient enterohepatic recirculation, metabolism, and the homeostatic feedback mechanisms connecting hepatocytes and enterocytes. Importantly, also the luminal gut microbiome is known to modulate bile acid composition via deconjugation of the primary bile acids. Changes in the intestinal microbiome of CARD14(E138A)^{IEC} mice could therefore affect epithelial-neuronal communication by altering the gut bile acid composition. Given the complexity and redundancy of gut hormone and bile acid signaling, and their bidirectional relationship with the gut microbiota, it will be very difficult to pinpoint the specific role of each in the regulation of intestinal motility in CARD14(E138A)^{IEC} mice.”

Specific points:

1. The authors should show Paneth cell dysfunction is induced by TNF by introducing TNF deficiency into CARD14 (E138A)^{IEC} mice.

RESPONSE: As also explained in our response to reviewer 1, we now studied the role of TNF in intestinal motility using two different approaches. Treatment with TNF neutralizing antibodies did not prevent the CARD14(E138A)-induced delay in intestinal transit. The above has now been added in the results (page 10) and discussion (page 13) section of the revised manuscript and is shown as Fig. 6J. In addition, we now also analyzed intestinal transit time in TNF^{emARE} mice, which overexpress TNF and spontaneously develop intestinal inflammation. TNF^{emARE} mice and WT control mice showed similar intestinal motility (Fig. 6K in the revised manuscript). Together, these data indicate that TNF is not a central driver of delayed intestinal motility in CARD14(E138A)^{IEC} mice, and that other mechanisms likely contribute to Paneth cell dysfunction and impaired motility.

2. The authors should also show whether the inflammation causes the delayed intestinal motility. In Fig.3, the authors show that MLT827 did not recover the intestinal motility in the mutant mice. In this case, was intestinal inflammation not cancelled?

RESPONSE: Although we cannot fully exclude a causal role for inflammation in delayed intestinal motility in our mouse model, several experiments (neutrophil depletion, TNF neutralisation, absence of neuronal damage; as already discussed above) make this rather unlikely. We did not analyse the effect of MALT1 inhibitor treatment on CARD14(E138A)-induced neutrophilia in our original experiments shown in the manuscript and did not get ethical approval for another experiment. However, we have now also analysed proinflammatory gene expression in IEC from control and CARD14(E138A)^{IEC} mice treated or not with the MALT1 inhibitor in the original experiments. Although, MALT1 inhibitor treatment clearly suppressed MALT1 activity as evidenced by the inhibition of the CARD14(E138A)-induced CYLD cleavage (figure 3D), CARD14(E138A)-induced expression of *Tnf* and *Cxcl2* were unaffected (new Figure 3E). These data suggest that MALT1 proteolytic activity does not have a major contribution to the inflammation effect of CARD14(E138A) in the intestine. It should be mentioned that MALT1 catalytic activity is known to only affect the expression of a limited number of genes in activated T cells, most likely reflecting the cleavage of RNA destabilizing proteins, such as Roquin. We can therefore not fully exclude a similar effect of MALT1 catalytic activity on the expression of other genes in IEC.

3. The authors should also show the mechanisms into how epithelial-neuronal communication is induced. Is this inflammation-dependent?

RESPONSE: As outlined above in our response to the general comments of reviewer 3, and as suggested by reviewer 1, we have addressed this in the discussion of our revised manuscript and also extended this with a potential role for bile acid metabolism.

4. In the final part of Discussion, the authors stated that patients with the gain-of-function of CARD14 mutation do not show intestinal inflammation. This reduces the value of this study.

RESPONSE: We presume the reviewer misinterpreted our wording as it was not our intention to make this statement and conclusion. Instead, we believe that our data have a high translational value. Several rare CARD14 variants, including CARD14(E138A), are associated with psoriasis and pityriasis rubra pilaris (a psoriasis-like skin inflammation) in human patients. Previously, we and others further validated this association using transgenic mice expressing the corresponding human CARD4 variants. Our new finding that human CARD14(E138A) expression in mice not only induces a skin phenotype but also a clear intestinal phenotype, strongly indicates the existence of a similar intestinal phenotype in humans carrying the CARD14(E138A) variant. That this has not yet been reported in such patients does not mean it does not occur, but most likely reflects the fact that CARD14(E138A) and other variants leading to psoriasis are rare genetic variants as well as the fact that intestinal discomfort is not part of the dermatology patient questionnaire or something that a patient brings up during a visit at a dermatologist. Moreover, the high frequency of constipation in the general human population (1/6), combined with the rare nature of CARD14 variants, makes it very challenging to demonstrate a causative role of CARD14 variants in intestinal dysmotility in human patients. It would require a very high number of patients with CARD14 variants and a large multi-center study, which goes far beyond the scope of our study. Nevertheless, our study sends the strong message that it might be important to increase awareness for intestinal disease symptoms among certain psoriasis patients and treating dermatologists. Therefore, we consider our study as highly relevant for human health.

The above was also mentioned in the discussion section of our original manuscript (repeated below): *'Mild intestinal inflammation and delayed motility have not yet been reported in human psoriasis patients with hyperactivating CARD14 mutations but may have stayed unnoticed or neglected. Given the high frequency of constipation in the general human population (1/6)⁷⁴, and the fact that CARD14 mutations only account for a very small minority of cases (frequency of specific variants is ~0.013 or less⁴), demonstrating a causative role of CARD14 hyperactivation in intestinal dysmotility in human patients will also be very challenging. Nevertheless, our data indicate that it might be worth to increase awareness for intestinal disease symptoms among psoriasis patients and treating dermatologists.'*

Furthermore, a correlation between psoriasis and subclinical intestinal disease is also supported by a number of previously reported data in literature (as also mentioned by reviewer 2), showing that psoriasis is associated with low-grade intestinal inflammation, which may contribute to abdominal symptoms in patients and possibly set the stage for the development of intestinal disease (Lundquist et al., *Biochim Biophys Acta Mol Basis Dis* 1871(3):167634, 2025). We are now also referring to this study in the discussion of our revised manuscript and also add two other studies (PMID: 9217818 and PMID: 7066192) that support a link between psoriasis and intestinal disease.

In addition, as suitable genetic models to study intestinal motility disorders are currently lacking, the mouse model that we developed should be an asset for future studies that are relevant to multiple human diseases (e.g. constipation, IBS) characterized by changes in gut motility (as also indicated by reviewer 2), again illustrating the relevance to human health.

15th Sep 2025

Dear Dr. Afonina,

Thank you for submitting your revised study. We have now received the reports from referees #1 and #3 who were asked to review your revised manuscript. As you will see below, they are overall satisfied with the revisions, and I will therefore be able to accept your manuscript once the following editorial concerns are addressed:

1/ Manuscript text:

- Please remove the yellow highlights and only keep in track changes mode any new modification in the text.
- We can accommodate a maximum of 5 keywords, please adjust accordingly.
- Please remove the list of abbreviations, and instead define each abbreviation the first time it appears in the text.
- Materials and Methods should be renamed Methods:
 - o Please indicate the origin of all mice (including in the Reagents and Tools table).
 - o Please provide dilutions/concentrations for all antibodies.
- Data availability section: Please provide a URL for each deposited dataset. Please note that the datasets must be publicly available before acceptance of the manuscript.
- Thank you for providing The Paper Explained. Please add it to the manuscript text file.

2/ Figures:

- Tables EV1 and EV2 should be uploaded as separate files and renamed Dataset EV1 and EV2.
- Please address the queries from our data editors in the figure legends:
 1. Please note that the exact p values are not provided in the legends of figures 2B, D, E; 3A, D.
 2. Please indicate the statistical test used for data analysis in the legends of figures 5A, B, C; 6J, EV2 A, F.
 3. Please note that the box plots need to be defined in terms of minima, maxima, centre, bounds of box and whiskers, and percentile in the legend of figure 7A.

3/ Source Data: Please provide the source data for the main manuscript figures as described in your checklist.

4/ Author checklist: please fill in "Experimental study design and statistics/ inclusion-exclusion criteria".

5/ Thank you for providing a nice visual abstract. Please resize to a png/tiff/jpeg file 550 px wide x 300-600 px high. A cropped portion of this image will serve as thumbnail for the table of content on our webpage.

6/ As part of the EMBO Publications transparent editorial process initiative (see our Editorial at <http://embomolmed.embopress.org/content/2/9/329>), EMBO Molecular Medicine will publish online a Review Process File (RPF) to accompany accepted manuscripts.

This file will be published in conjunction with your paper and will include the anonymous referee reports, your point-by-point response and all pertinent correspondence relating to the manuscript. Let us know whether you agree with the publication of the RPF.

I look forward to receiving your revised manuscript.

Yours sincerely,

Lise Roth

***** Reviewer's comments *****

Referee #1 (Comments on Novelty/Model System for Author):

The manuscript demonstrates high technical quality, with appropriate methodology and robust statistical analysis that support the validity of the findings. The novelty of the study is strong, offering original insights that advance the field. While the medical impact is currently of medium level, the results provide a solid foundation for future translational work that could enhance clinical relevance. The model system employed is adequate for the scope of the study, and the conclusions are well supported by the data generated.

Referee #1 (Remarks for Author):

The authors have carefully addressed all the points raised in the previous review. The revisions made to the manuscript are satisfactory and have significantly improved the clarity, rigor, and overall quality of the work. The study remains original and impactful, with strong methodological rigor and well-presented results. The discussion is balanced and integrates the findings effectively into the broader context of the field.

Referee #3 (Remarks for Author):

The authors responded to the reviewer's comments, and the manuscript is now somewhat improved.

The authors addressed the remaining editorial issues.

29th Sep 2025

Dear Dr. Afonina,

Thank you for submitting your revised files. I am pleased to inform you that your manuscript is accepted for publication and is now being sent to our publisher to be included in the next available issue of EMBO Molecular Medicine!

Yours sincerely,

Lise Roth
